# Learning Dynamics in Continual Pre-Training for Large Language Models

**Xingjin Wang** [1 2]   **Howe Tissue** ✉   **Lu Wang** [3]   **Linjing Li** [1 2]   **Daniel Dajun Zeng** [1 2]

## Abstract

Continual Pre-Training (CPT) is a popular and effective method for applying strong foundation models to specific downstream tasks. In this work, we explore the *learning dynamics* throughout the CPT process for large language models. We specifically focus on how general and downstream domain performance evolves at each training step, with performance measured by validation losses. We observe that the CPT loss curve fundamentally characterizes a transition from an initial pre-training trajectory to a new, domain-specific one, conceptualized as a shift between two hidden loss curves. This transition can be described by decoupling the effects of distribution shift and learning rate annealing. We derive a CPT scaling law that combines these two factors, enabling the prediction of loss at any (continual) training step and across various learning rate schedules. Our formulation presents a comprehensive understanding of several critical factors in CPT, including loss potential, peak learning rate, training steps, and replay ratio. Moreover, our approach can be adapted to optimize training hyper-parameters for different CPT goals, such as balancing general and domain-specific performance. Extensive experiments demonstrate that our scaling law holds across various CPT datasets and hyper-parameters.

## 1. Introduction

In recent years, large language models (LLMs) have exhibited versatile abilities and garnered significant academic and industrial attention (Dubey et al., 2024; OpenAI, 2023).

[1]School of Artificial Intelligence, University of Chinese Academy of Sciences, Beijing, China [2]State Key Laboratory of Multimodal Artificial Intelligence Systems, Institute of Automation, Chinese Academy of Sciences, Beijing, China [3]Ritzz-AI. `lwzzfzl@gmail.com`. Correspondence to: Howe Tissue (project lead) <`h-sun20@tsinghua.org.cn`>.

*Proceedings of the 42$^{nd}$ International Conference on Machine Learning*, Vancouver, Canada. PMLR 267, 2025. Copyright 2025 by the author(s).

Continual Pre-Training (CPT) of LLMs aims to enhance their abilities in specific downstream domains (e.g. coding, finance, math) while mitigating the substantial costs associated with re-training (Chen et al., 2023a; Çağatay Yıldız et al., 2024; Ibrahim et al., 2024).

CPT primarily involves a trade-off between performance on general and downstream domains. It is widely observed that improvements on downstream tasks may come at the expense of degrading performance on general domain tasks, a phenomenon known as catastrophic forgetting (French, 1999; Gupta et al., 2023). Recently, some scaling laws have been proposed for CPT scenarios. For example, Hernandez et al. (2021b) and Barnett (2024) discovered a law describing how data transfer effectiveness scales with fine-tuning dataset size and model size. Que et al. (2024) and Gu et al. (2024) proposed a law to find the optimal replay ratio to balance general and downstream performances.

However, very few studies have attempted to *quantitatively* describe the *learning dynamics* of CPT, particularly how performance varies on general and downstream domains throughout the CPT process. We have two primary research questions (RQs): (1) *Can we derive an accurate law describing the influence of as many variables as possible on the final CPT performance?* (2) *Can we trace the performance of LLMs throughout the entire CPT process, rather than only the final performance?* Studying the first RQ will help researchers investigate various factors that affect CPT performance and facilitate hyper-parameters optimization through prediction; studying the second RQ will help the community understand the learning dynamics of LLMs at each step of the CPT process, providing deeper insights and theoretical guidance for subsequent CPT research.

Following previous works (Gupta et al., 2023; Ibrahim et al., 2024; Que et al., 2024), we trace performance changes using validation losses on corresponding domains. We find that the CPT loss curve acts as a transfer curve and can be described by decoupling the effects of *distribution shift* and *learning rate (LR) annealing*. Specifically, the distribution shift between the pre-training (PT) and CPT data leads to a deviation in the loss curve, while LR annealing results in a loss decrease in both the PT and CPT phases. By analyzing various loss curves, we discover a CPT scaling law that integrates these two factors, enabling accurate prediction of

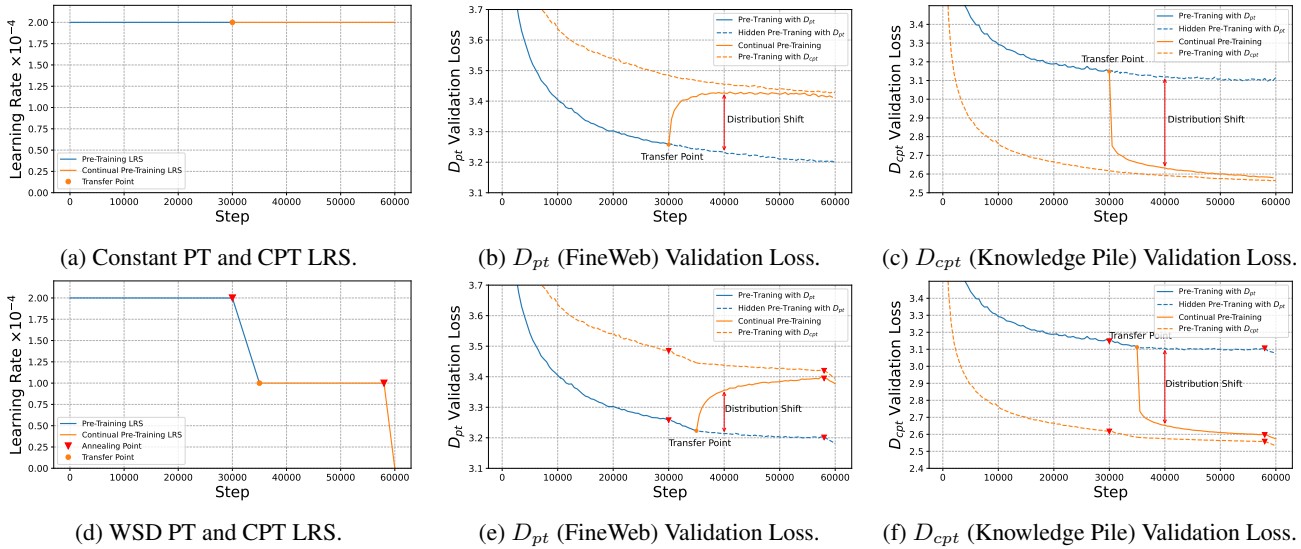

(a) Constant PT and CPT LRS.  (b) $D_{pt}$ (FineWeb) Validation Loss.  (c) $D_{cpt}$ (Knowledge Pile) Validation Loss.

(d) WSD PT and CPT LRS.  (e) $D_{pt}$ (FineWeb) Validation Loss.  (f) $D_{cpt}$ (Knowledge Pile) Validation Loss.

*Figure 1.* CPT loss curves under different learning rate schedules (LRS): constant (a-c) and warmup-stable-decay (WSD) (Hu et al., 2024) (d-f). The CPT loss curve acts as a transfer curve from the hidden PT curve trained on $D_{pt}$ (Blue dashed) to the hidden PT curve trained on $D_{cpt}$ (Orange dashed). The transfer curve converges to the hidden PT curve trained on $D_{cpt}$.

losses throughout the entire CPT phase.

Our proposed scaling law provides a comprehensive model of how key variables affect the training dynamics of CPT, such as *loss potential* (defined in section 3.3), peak LR, training steps, and replay ratio. We demonstrate how these variables jointly affect model performance at each CPT step, and how to optimize these hyper-parameters for better CPT performance. By applying our scaling law, several valuable conclusions emerge. For example: (1) PT models with higher loss potential can better adapt to downstream domains in CPT; (2) The performance degradation on the PT domain during the CPT phase is inevitable if the *turning length* is infinitely large, which implies that the PT model is adequately trained or the distribution shift between the PT and CPT data is very large; (3) For specific CPT goals, like balancing performance between the PT and CPT domains, or optimizing out-of-domain performance, our scaling law can predict the optimal training hyper-parameters such as the loss potential, peak LR, and PT dataset replay ratio.

## 2. Pilot Observation

### 2.1. Task Formulation

We investigate the dynamics of performance in both general and downstream domains during the CPT process. Following previous works (Ibrahim et al., 2024; Que et al., 2024; Gu et al., 2024; Hernandez et al., 2021a), we assess model performance by examining the validation loss on the PT dataset $D_{pt}$ and the CPT dataset $D_{cpt}$.

**Experimental Setup.** Our main experiments employ LLaMA-like models (Dubey et al., 2024) with 106M to 1.7B non-embedding parameters. We use FineWeb (Penedo et al., 2024) as $D_{pt}$ and Knowledge-Pile (Fei et al., 2024) as $D_{cpt}$. We leverage different LRS in the PT and CPT phases (see Fig. 1). More details are provided in Appendix B.

**Observation.** As observed in previous studies (Ibrahim et al., 2024; Gupta et al., 2023), during the CPT process, the $D_{pt}$ validation loss tends to increase (Fig. 1b and Fig. 1e), whereas the $D_{cpt}$ validation loss decreases (Fig. 1c and Fig. 1f). Moreover, in both PT and CPT phases, the loss curve is significantly influenced by the LRS. For example, the loss decreases rapidly when the LR anneals, which is observed in our prior work (Tissue et al., 2024).

### 2.2. CPT Transfer Loss Curve

To enhance our understanding of the CPT training dynamics, we train two additional loss curves: the hidden PT curve trained on $D_{pt}$ and the hidden PT curve trained on $D_{cpt}$.

**Hidden PT Curve trained on $D_{pt}$.** This curve represents the loss when the model is consistently pre-trained using $D_{pt}$ with the same LRS as used in the CPT phase.

**Hidden PT Curve trained on $D_{cpt}$.** This curve depicts the loss when the model is trained from scratch on $D_{cpt}$, while adhering to the same training setups (such as LRS) as those applied in the PT and CPT phases.

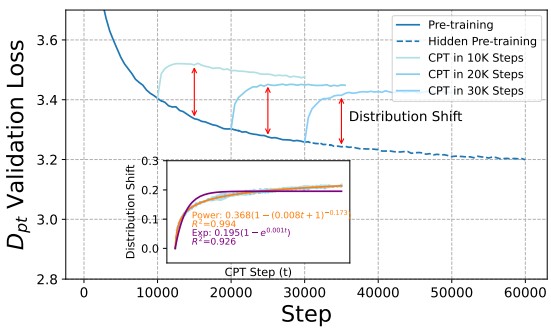

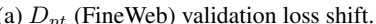

(a) $D_{pt}$ (FineWeb) validation loss shift.

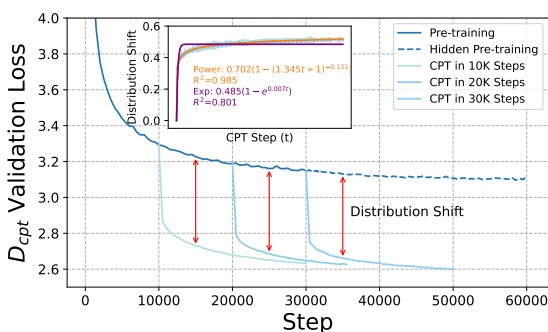

(b) $D_{cpt}$ (Knowledge Pile) validation loss shift.

*Figure 2.* The transfer loss curve in $D_{pt}$ and $D_{cpt}$ validation sets for different transfer starting points with constant LRS. We find that the distribution shift term is independent of the transfer starting points and adheres to a power-law form.

**Transfer Curve.** As shown in Fig. 1, the CPT loss curve acts as a ***transfer curve*** between these two hidden PT curves; i.e., the CPT loss deviates from the hidden PT curve trained on $D_{pt}$ and converges towards the hidden PT curve trained on $D_{cpt}$. The discrepancy between the transfer loss curve and the hidden PT curve trained on $D_{pt}$ is called *distribution shift*. As the number of CPT steps approaches infinity, the CPT loss is expected to converge to the hidden PT curve trained on $D_{cpt}$.

> **Finding 1.** The process of CPT is how the loss curve transitions from the hidden PT curve trained on $D_{pt}$ to the hidden PT curve trained on $D_{cpt}$.

## 3. Continual Learning Dynamics Law

We quantitatively analyze the transfer curve by modeling the effects of LR annealing and distribution shift.

### 3.1. LR Annealing

Without data transfer, the CPT loss curve would follow the trajectory of the hidden PT curve trained on $D_{pt}$. Tissue et al. (2024) introduced a scaling law to describe the loss dynamics at each step $t$ as affected by LR annealing:

$$L(t) = L_0 + A \cdot S_1^{-\alpha} - C \cdot S_2, \quad (1)$$

where the forward area $S_1 = \sum_{i=1}^{t} \eta_i$ is the summed LR, and the annealing area $S_2 = \sum_{i=1}^{t} \sum_{k=1}^{i} (\eta_{k-1} - \eta_k) \cdot \lambda^{i-k}$ is a term affected by LR annealing. $L_0, A, C, \alpha$ are constant positive parameters to be fitted. $\lambda = 0.999$ is a hyper-parameter related to the momentum term.

The loss in the CPT process without distribution shift (denoted as $L_{base}(t)$) follows this law, i.e.,

$$L_{base}(t) = L_0 + A \cdot (S_1^{pt} + S_1^{cpt})^{-\alpha} - C \cdot (S_2^{pt} + S_2^{cpt}), \quad (2)$$

where $t$ denotes the CPT step, and $S_1^{pt}(S_2^{pt})$ and $S_1^{cpt}(S_2^{cpt})$ are the forward (annealing) areas at the PT and CPT stages, respectively.

### 3.2. Distribution Shift Term

The distribution shift term describes the deviations from the hidden PT curve trained on $D_{pt}$. This shift reflects the distributional distance between $D_{pt}$ and $D_{cpt}$. Many studies (Ibrahim et al., 2024; Wang et al., 2024; Parmar et al., 2024) have highlighted the impact of LRS at the CPT stage, implying that this shift should be also affected by the LRS. We first analyze the form of the distribution shift term with a constant LR to isolate the effects of LRS, then we incorporate the forward area into the equation to accurately describe the distribution shift term for different LRS.

**Constant LRS.** We first use a constant LR in both PT and CPT phases. To study the relationship between distribution shift and the PT model state, we continually pre-train the model starting from different transfer points. As shown in Fig. 2, these distribution shift terms tend to overlap regardless of the transfer starting point. This overlap suggests that *the distribution shift term is independent of transfer starting points or PT model checkpoints.*

We compare to fit the distribution shift term using exponential and power-law forms, and find the best fit to be $\Delta L(t) = B \cdot (1 - (E \cdot t + 1)^{-\beta})$. We do not adopt the simple power-law form $\Delta L(t) = B \cdot t^{-\beta}$ to ensure that $\Delta L(0) = 0$. We leverage this equation to fit the transfer loss curve of both $D_{pt}$ and $D_{cpt}$ validation sets, as shown in Fig. 2.

**Other LRS.** When considering the effect of LRS, we find that the LR values, i.e., the forward area in Eq. 1, significantly affects the distribution shift term. The smaller forward area in the CPT results in a smaller distribution

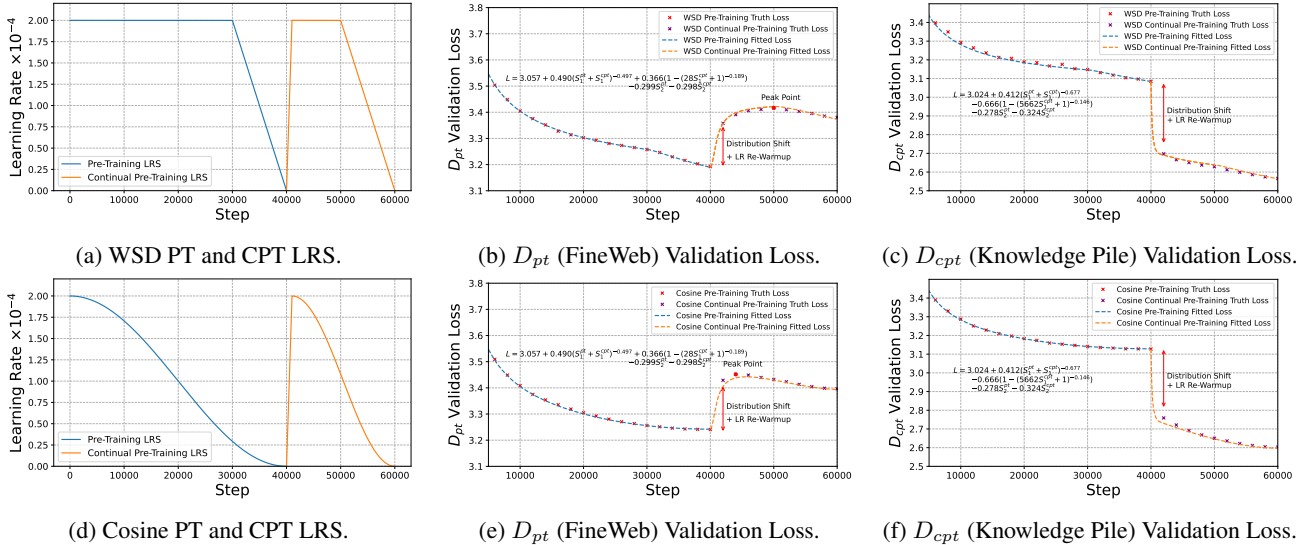

(a) WSD PT and CPT LRS.    (b) $D_{pt}$ (FineWeb) Validation Loss.    (c) $D_{cpt}$ (Knowledge Pile) Validation Loss.

(d) Cosine PT and CPT LRS.    (e) $D_{pt}$ (FineWeb) Validation Loss.    (f) $D_{cpt}$ (Knowledge Pile) Validation Loss.

*Figure 3.* Using Eq. 4 to fit all PT and CPT loss curves with different LRS (WSD and Cosine). For $D_{pt}$ validation sets, all loss curves (**b** and **e**) are described by the same equation; similarly, for $D_{cpt}$ validation sets, all loss curves (**c** and **f**) follow the same equation.

shift, as shown in different transfer curves in Fig. 1b vs. 1e (or Fig. 1c vs. 1f). Hence, following Tissue et al. (2024), we replace the training steps $t$ with the forward area $S_1^{cpt}$ in the CPT phase:

$$\Delta L(t) = B \cdot (1 - (1 + E \cdot S_1^{cpt})^{-\beta}), \quad (3)$$

which instead adopts $S_1^{cpt}$ to represent the training amount in CPT stage, considering the impact of LR values.

### 3.3. Final Transfer Curve

We combine the effect of LR annealing (Eq. 2) and distribution shift (Eq. 3) to get the equation for the CPT loss:

$$L(t) = L_{base}(t) + \Delta L(t)$$
$$= \underbrace{L_0 + A \cdot \left(S_1^{pt} + S_1^{cpt}\right)^{-\alpha} - C_1 \cdot S_2^{pt} - C_2 \cdot S_2^{cpt}}_{\text{Scaling law with LR annealing}} \quad (4)$$
$$+ \underbrace{B \cdot \left(1 - \left(1 + E \cdot S_1^{cpt}\right)^{-\beta}\right)}_{\text{Power-law distribution shift}}$$

We adopt different coefficients $C_1$ and $C_2$ for $S_2^{pt}$ and $S_2^{cpt}$ because the distributions of $D_{pt}$ and $D_{cpt}$ are different, and thus result in different annealing effects.

Our equation can predict the loss at any step with any LRS during both the PT and CPT phases. We conduct experiments utilizing the widely adopted WSD (Hu et al., 2024) and cosine (Loshchilov & Hutter, 2016) LRS in the PT and CPT phases (see Fig. 3a and Fig. 3d). We use Eq. 4 to fit all loss curves on the $D_{pt}$ and $D_{cpt}$ validation sets. As illustrated in the middle and right panels of Fig. 3, our equation successfully captures the trends in loss variations

across different LRS throughout the training process. We also use the fitted equation to predict loss curves of other LRS, and the prediction accurately matches the observation (see Fig. 10). Furthermore, the batch size and sequence length may change in the CPT phase. However, our scaling law equation remains adaptable to these hyper-parameter changes, as demonstrated in Appendix F.

> **Finding 2.** The CPT loss curve can be decomposed into a hidden PT curve trained on $D_{pt}$ and a distribution shift term. The hidden PT curve trained on $D_{pt}$ is formalized as a scaling law with LR annealing, whereas the distribution shift term is independent of transfer starting points and adheres to a power-law form.

**Transfer Loss Surface.** To better understand our formulation, we follow Tissue et al. (2024) to view the loss surface of LLMs as a *slide*-like transition between surfaces in Fig. 4. The CPT process transitions from one surface to another following a power-law form. A larger distributional distance between $D_{pt}$ and $D_{cpt}$ leads to a steeper slope of the transfer surface, and thus a sharper increase in the $D_{pt}$ loss. When the LR anneals, the amplitude of the oscillation on the loss surface decreases, and thus the loss also decreases. In the annealing view, we term the "height" of the current model state as its *loss potential*. We use this concept to capture the potential for future loss drop via LR annealing. Quantitatively, we can define loss potential as the ratio of the final annealed LR of the PT phase to the peak learning rate in the PT phase.

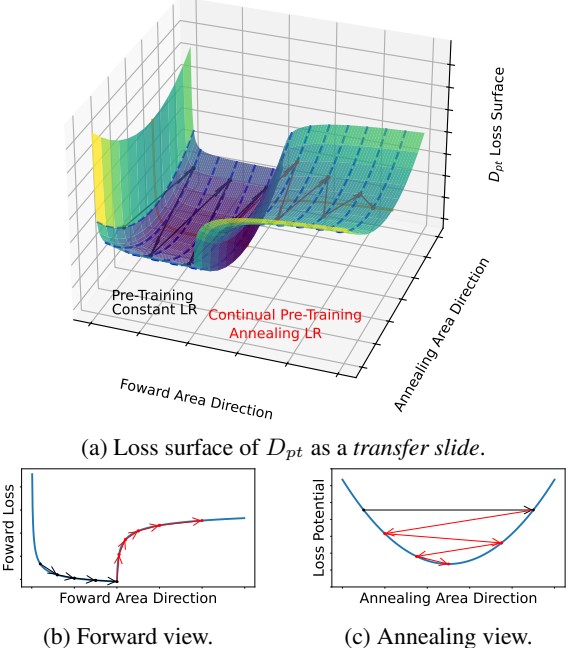

(a) Loss surface of $D_{pt}$ as a *transfer slide*.

(b) Forward view.          (c) Annealing view.

*Figure 4.* The loss surface of the CPT process and two directional views.

## 3.4. Extension to Model Size and Replay Ratio

We attempt to incorporate model size $N$ into our CPT scaling law by analyzing the effect of $N$ on both the LR annealing and distribution shift term. Our experiments show that the distribution shift terms remains unchanged across different model sizes when other settings are fixed (see details in Appendix E). Therefore, we can directly follow Tissue et al. (2024) to integrate an $N$-related term and use our scaling law to fit and predict CPT loss curves for different model sizes. More discussion is provided in Appendix E.

We also integrate the replay ratio into our scaling law since replaying some data from $D_{pt}$ is a common practice in CPT. Our experiments show that the replay ratio influences the distribution shift term in an *exponential* manner. By adding a single replay ratio related term, our scaling law can predict the entire training dynamic for different replay ratios, while previous studies (Que et al., 2024) can only predict the final loss. More details are given in Appendix H.

## 4. Factor Analyses and Applications

In this section, we analyze various factors for CPT and apply our scaling law to provide insights into these factors.

## 4.1. Loss Potential

Most PT models are trained by annealing to a minimum LR for lower PT losses. However, the optimal PT model for CPT is not necessarily a fully annealed model. We use the concept of loss potential introduced in section 3.3 to describe the degree of annealing for PT models. Specifically, a PT model trained without annealing has a high loss potential, while a PT model that anneals to a zero LR value has a low loss potential. We investigate the impact of loss potential on CPT under two different experimental settings: without or with LR re-warmup.

**W/o Re-warmup.**    In this setting, we set the initial LR for CPT as the final LR in PT and linearly anneal the LR for CPT to zero. We conduct experiments using PT models with different loss potentials (Fig. 5a). As shown in Fig. 5b, models with higher loss potential achieve lower final losses on $D_{cpt}$. This observation matches the prediction made by our CPT scaling law (Fig. 5c). We also utilize our equation to predict the final loss across various CPT steps, confirming that this trend persists in different settings.

**With Re-warmup.**    A common practice for CPT is to linearly re-warmup the LR from zero to a certain value, such as 10% of the peak LR in PT, before annealing it to zero (Fig. 5d). As shown in Fig. 5e and Fig. 5f, models with high loss potential consistently achieve lower final losses.

We can use our CPT scaling law (Eq. 4) to analyze the impact of loss potentials. Specifically, as the annealing coefficient $C_2 > C_1$ often holds for $D_{cpt}$, then allocating a larger annealing area in the CPT phase, i.e., a larger $S_2^{cpt}$, facilitates a lower loss. Moreover, models with higher loss potential have larger forward areas $S_1^{pt}$ and $S_1^{cpt}$, which further contribute to a lower loss. Therefore, PT models with high loss potential usually lead to lower $D_{cpt}$ loss. This conclusion is also validated in previous works (Wang et al., 2024).

> **Finding 3.** PT models with higher loss potential consistently achieve lower $D_{cpt}$ validation losses. Hence, we advocate that *when releasing open-source models, it is beneficial to release a high loss potential version to facilitate downstream tasks.*

## 4.2. Replay Ratio

The distributional distance between $D_{pt}$ and $D_{cpt}$ significantly influences the distribution shift term in Eq. 4. As shown in Fig. 6a, a more distinct $D_{cpt}$, Pile of Law (Henderson* et al., 2022), leads to a sharper transfer curve than a more similar $D_{cpt}$ (Knowledge Pile (Fei et al., 2024)).

In CPT, it is a common practice to mix $D_{pt}$ into $D_{cpt}$ based on a certain *replay ratio* to mitigate the increase of validation loss on $D_{pt}$. The replay ratio plays a critical role in adjusting the distributional distance between $D_{pt}$ and $D_{cpt}$ since $D_{cpt}$

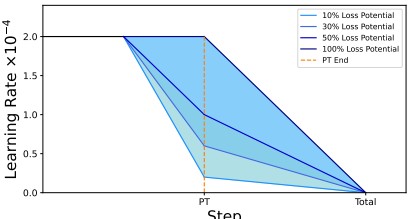

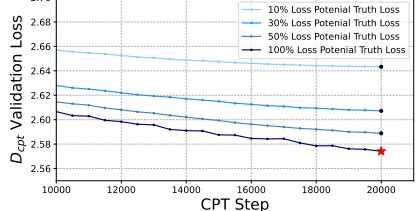

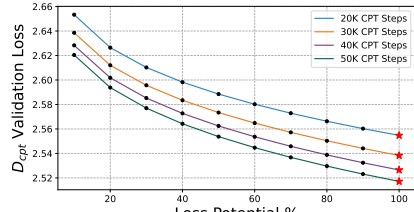

(a) CPT with different loss potentials (w/o re-warmup setting).

(b) $D_{cpt}$ true loss vs. CPT step for different loss potentials (w/o re-warmup setting).

(c) $D_{cpt}$ predicted loss vs. loss potentials for different CPT steps (w/o re-warmup setting).

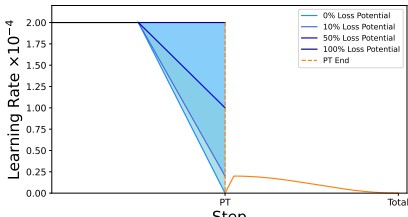

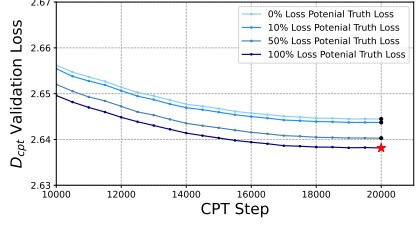

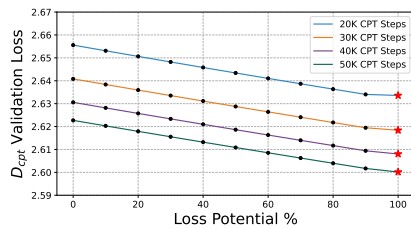

(d) CPT with different loss potentials (w/ re-warmup).

(e) $D_{cpt}$ true loss vs. CPT step for different loss potentials (w/ re-warmup setting) .

(f) $D_{cpt}$ predicted loss vs. loss potentials for different CPT steps (w/ re-warmup setting).

*Figure 5.* The impact of the loss potential of PT models. We illustrate the true loss of models with different loss potential in the middle panel. We utilize Eq. 4 to predict the losses of these models across different training steps in the right panel. The red star (⋆) refers to the models that achieve the lowest $D_{cpt}$ validation loss given the number of CPT steps.

is modified to approach $D_{pt}$ [1]. Results in Fig. 6b and Fig. 6c indicate that higher replay ratios lead to smaller distribution shifts and thus effectively decelerate the deviation from $D_{pt}$. Quantitatively, we find that the replay ratio influences the distribution shift term based on an *exponential* form, which is elaborated in Appendix H.

### 4.3. Peak LR

In real scenarios, choosing an appropriate peak LR for re-warmup is important for CPT. Different peak LRs affect the $D_{pt}$ and $D_{cpt}$ validation loss. We leverage Eq. 4 to predict the final loss of different peak LRs. Specifically, we assume the PT model is trained using the WSD LRS. As shown in Fig. 7a and Fig. 7b, a high peak LR in the CPT phase accelerates the decrease of the $D_{cpt}$ validation loss while leading to an increase of the $D_{pt}$ validation loss.

### 4.4. CPT Training Steps

The number of CPT training steps is also an important hyper-parameter. A general observation is that more training steps lead to lower $D_{cpt}$ validation loss. However, the $D_{pt}$ validation loss may exhibit three different patterns based on the state of the PT model and the distributional distance between $D_{pt}$ and $D_{cpt}$: (1) a continuous rise; (2) an initial

rise followed by a decline that does not return to the original loss value; or (3) an initial rise followed by a decline that goes below the original loss value.

As shown in Fig. 7c, we define the *critical point* (indicated by the blue dashed line) as the convergence value of the $D_{pt}$ loss on the hidden PT curve trained on $D_{cpt}$. When CPT occurs before this critical point, the $D_{pt}$ loss will first rise and then decline. The final loss may or may not be lower than the original loss. The minimum training steps required to return to the initial loss value are designated as the ***turning length***. Conversely, if CPT occurs after the critical point, achieving a lower $D_{pt}$ loss than the initial value becomes unattainable, regardless of how many steps we train.

> **Finding 4.** Inadequate pre-training or weak distribution shift can result in lower $D_{pt}$ loss values after sufficient CPT steps compared to the PT model. Otherwise, we are unlikely to achieve a lower $D_{pt}$ loss than the PT model, regardless of how many CPT steps we train. In this situation, more training often leads to degraded general performance.

## 5. Balance Between $D_{pt}$ and $D_{cpt}$ Loss

Validation losses on $D_{pt}$ and $D_{cpt}$ typically exhibit a trade-off in the CPT process. Balancing these losses is critical for optimizing the overall performance of the model during CPT. We define the increase in $D_{pt}$ loss as $\Delta L_{D_{pt}}$ and the

---

[1]It is important to note that the $D_{cpt}$ undergoes modifications in the presence of replays. For instance, if we mix 0.1 FineWeb with 0.9 KP, the actual $D_{cpt}$ distribution is 0.1 FineWeb with 0.9 KP, not 1.0 KP.

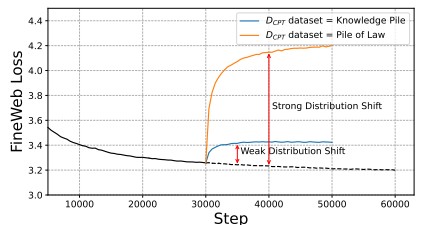 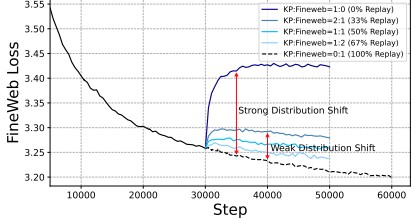 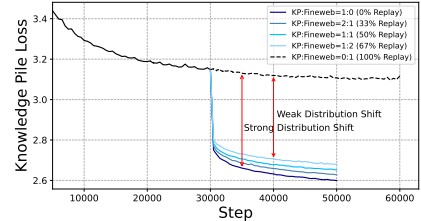

(a) The difference in distribution shift for different $D_{cpt}$ datasets.

(b) The distribution shift on the $D_{pt}$ validation set for different replay ratios.

(c) The distribution shift on the $D_{cpt}$ validation set for different replay ratios.

*Figure 6.* We compare the distribution shift for different distributional distances between the $D_{cpt}$ and $D_{pt}$ datasets. Additionally, we examine the impact of different replay ratios on the distribution shifts within both the $D_{cpt}$ and $D_{pt}$ validation sets.

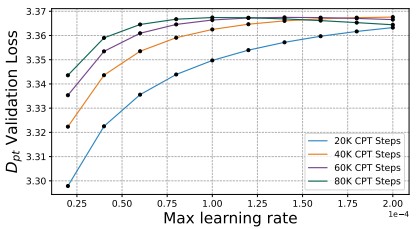 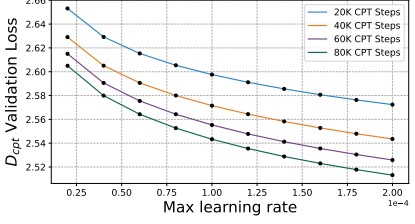 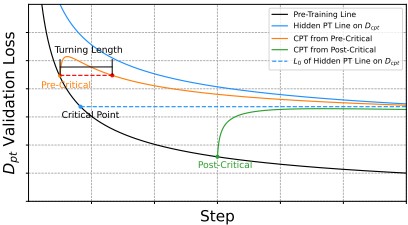

(a) $D_{pt}$ predicted loss vs. peak LRs for different CPT steps.

(b) $D_{cpt}$ predicted loss vs. peak LRs for different CPT steps.

(c) Critical point and turning length in $D_{pt}$ validation loss.

*Figure 7.* **(a)-(b) The effect of the peak LR.** We utilize Eq. 4 to predict the final loss for different peak LRs. **(c) The effect of different CPT steps.** We show the critical point and turning length in the $D_{pt}$ validation loss.

decrease in $D_{cpt}$ loss as $\Delta L_{D_{cpt}}$. To balance the loss of $D_{pt}$ and $D_{cpt}$ validation sets, we assign normalized balance coefficient to different validation sets:

$$\min_{S_1^{cpt}, S_2^{cpt}} \quad \lambda_1 \Delta L_{D_{pt}} + \lambda_2 \Delta L_{D_{cpt}}$$
$$\text{s.t. } \lambda_1 + \lambda_2 = 1 \tag{5}$$

where $\lambda_1$ and $\lambda_2$ are coefficients that should be set based on our prior knowledge of the relative importance of general and downstream performance.

### 5.1. Optimal Hyper-Parameters

Given the different coefficients $\lambda_1$ and $\lambda_2$, there exist some optimal CPT hyper-parameters.

**Loss Potential.** Fig. 8a shows the optimal loss potential for different values of $\lambda_1$. It can be observed that a small $\lambda_1$ corresponds to a large optimal loss potential. This makes sense since a small $\lambda_1$ means that the final loss is dominated by the $D_{cpt}$ loss, and thus it is necessary to reserve sufficient loss potential for downstream domains.

**Peak LR.** We can also predict the optimal peak LR in the CPT process when $\lambda_1$ is given (Fig. 8b). A larger $\lambda_1$ suggests a preference for minimizing the increase in $D_{pt}$ loss, thereby necessitating a lower peak LR.

**Replay Ratio.** Based on our scaling law with replay ratio Eq. 8, we can determine the optimal replay ratio for each $\lambda_1$ (Fig. 8c). The same distribution line (dashed line) in Fig. 8c indicates that the optimal replay ratio should be the same as the target weight $\lambda_1$ if we initialize the CPT model randomly rather than from a pre-trained model. Instead, in practice, the optimal replay ratio shifts because the PT model has already been trained on $D_{pt}$, which causes the curve to deviate and exhibit a wave pattern.

**CPT Training Steps.** As shown in Fig. 13, we can get different turning lengths for different values of $\lambda_1$. When $\lambda_1$ is small, the $D_{cpt}$ loss predominates the composite loss $\lambda_1 L_{D_{pt}} + \lambda_2 L_{D_{cpt}}$, which consistently remains below the initial value. Conversely, with a moderate $\lambda_1$, there exists a specific step that makes the composite loss equals the inital loss. For a large $\lambda_1$, the composite loss is always higher than the initial loss, which means that CPT is not suitable any more in this situation.

### 5.2. Out-of-Domain Validation Set

Note that our CPT scaling law is designated to predict losses on $D_{pt}$ and $D_{cpt}$ validation sets, while it is not directly applicable to the out-of-domain (OOD) validation set $D_{ood}$. Inspired by previous works (Ye et al., 2024; Zhang et al., 2025) that the OOD validation loss can be represented as a

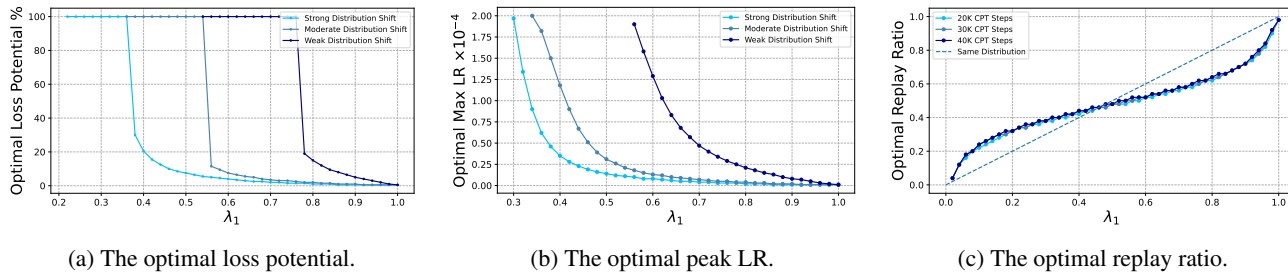

(a) The optimal loss potential.

(b) The optimal peak LR.

(c) The optimal replay ratio.

*Figure 8.* Optimizing hyper-parameters for CPT based on different coefficients to balance general and downstream performance. Strong, moderate, weak distribution shift in (a) and (b) denote different CPT datasets as Pile of Law, Knowledge Pile, and a mixture of 67% FineWeb and 33% Knowledge Pile, respectively. The "same distribution" in (c) represents a reference line where the target weight ($\lambda_1$) is the same as replay ratio. See Appendix I for more details.

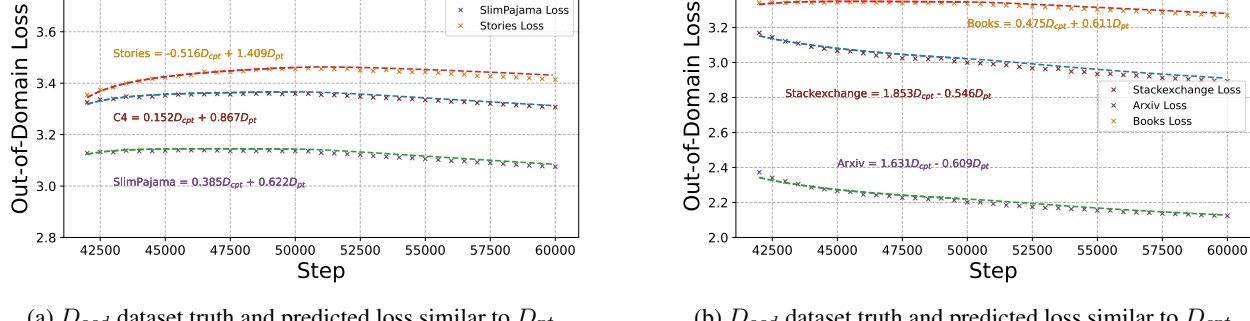

(a) $D_{ood}$ dataset truth and predicted loss similar to $D_{pt}$.

(b) $D_{ood}$ dataset truth and predicted loss similar to $D_{cpt}$.

*Figure 9.* The predicted loss curve of $D_{ood}$ validation set, predicted by leveraging a linear combination of $D_{pt}$ and $D_{cpt}$ validation losses. We show some different shapes of rising curves similar to $D_{pt}$ on the left and some different shapes of falling curves similar to $D_{cpt}$ on the right.

linear combination of losses on several base domains, we hypothesize that the loss on $D_{ood}$ can be represented by a linear combination based on $D_{pt}$ and $D_{cpt}$ validation losses:

$$L_{D_{ood}} = \lambda_1' L_{D_{pt}} + \lambda_2' L_{D_{cpt}} \qquad (6)$$

We verify this hypothesis and calculate $\lambda_1'$ and $\lambda_2'$ for several example OOD datasets in Appendix K. Note that the coefficients $\lambda_1'$ and $\lambda_2'$ are related only to datasets and not to other training hyper-parameters.

**Loss Prediction of $D_{ood}$.** The $D_{ood}$ validation loss does not adhere to the formulation described in Eq. 4. However, by calculating and specifying the coefficients $\lambda_1'$ and $\lambda_2'$, it becomes feasible to predict the $D_{ood}$ loss curve using a linear combination of the $D_{pt}$ and $D_{cpt}$ loss curves. It is interesting that this problem reduces to the balance between $D_{pt}$ and $D_{cpt}$ loss (Eq. 5). The optimal hyper-parameters such as LR and replay ratio for this setting have been adequately discussed in the previous section.

As shown in Fig. 9, we first calculate the coefficients in Eq. 6 for several OOD datasets and then predict the corresponding validation losses. The almost perfect prediction suggests

that our approach are quite effective and practical in real scenarios. Moreover, the calculated coefficient represents the "similarity" between OOD datasets and $D_{pt}$ or $D_{cpt}$. As Fig. 9 shows, there are two kinds of OOD datasets: (1) $D_{pt}$-like one (larger $\lambda_1'$) with loss curve upward, and (2) $D_{cpt}$-like one (larger $\lambda_2'$) with loss curve downward.

> **Finding 5.** There exists an optimal loss potential, peak LR and replay ratio designated to balance $D_{pt}$ and $D_{cpt}$ losses. Besides, the turning lengths vary depending on the different balance weights. Predicting $L_{D_{ood}}$ is equivalent to balancing $D_{pt}$ and $D_{cpt}$ losses by utilization of linear combination tricks.

## 6. Open-Source PT Models

For the majority of LLM communities, the PT models we use are usually not trained by ourselves, but from open-source models. Most training details are not reported for those open-source PT models, i.e., the distribution of $D_{pt}$, the loss potential, and the PT training hyper-parameters are usually unknown. This inhibits the direct application

of our CPT scaling law. To solve this issue, we propose the following methods to make our scaling law become applicable again.

(a) *Firstly,* for the unknown PT dataset distribution, some methods based on probing (Hayase et al., 2024) have been proposed. Instead, we simply utilize an open-source Common Crawl dataset as a **proxy** $D_{pt}$ to approximate the distribution of $D_{pt}$. (b) *Secondly*, when fitting our scaling law, we regard some variables as unknown parameters to fit. For example, we treat $S_1^{pt}$ as a parameter that requires fitting to be close to the undisclosed real $S_1^{pt}$. (c) *Thirdly*, as most open-source PT models anneal to a minimal LR to get a better performance nowadays, we assume all open-source models anneal their LR to zero when calculating $S_2^{cpt}$. Refer to Appendix G for more details.

To verify our solutions for open-source PT models, we continually pre-train LLaMA3.2-1B (Dubey et al., 2024) and select the RedPajama (Weber et al., 2024) dataset as an **proxy** $D_{pt}$. As Fig. 18 in Appendix G shows, the almost perfect fitting and prediction for CPT loss curve of LLaMA3.2-1B suggests the effectiveness of our proposed methods. Moreover, this result also indicates that our scaling law can be easily extended to CPT scenarios with unknown PT model information, demonstrating the superiority of our scaling law to capture the learning dynamics of CPT.

## 7. Discussion

**Laws Formulation.** The formulation of $S_2$ in Eq. 1 can have other forms. For example, $S_2$ could also be a multi-power form (Luo et al., 2025), which is proposed following the work of Tissue et al. (2024). We adopt the equation form in Eq. 1 because it has fewer parameters and it works more effectively in practice. We also compare some format variates including adding a LR-weighted coefficient and adding a power term to $S_2$ (see more details in Appendix J). The experiments show that all formats lead to similar results while our formulation has superiority in simplicity (i.e. fewer parameters).

**Laws Fitting.** In our experiments, we predominantly employ constant, cosine, and WSD LRS to fit data, which are widely used in practical applications. It is worth noting that many other LR schedules could be also modeled. To apply our scaling law, we use common LRS (e.g. constant and cosine) to train a few steps to collect loss values. After parameters are fitted based on these values, our scaling law is also capable of predicting the loss curve under other specialized LRS for much longer training durations. Our scaling law shares the similar idea of fitting cost conservation with Tissue et al. (2024), thanks to our scaling law being able to describe the whole dynamics in CPT rather than only final loss.

**Limitations.** One main limitation of our work is that our laws are primarily based on empirical analyses and experimental verifications. We acknowledge that there is a lack of rigorous theoretical analysis and proof because it is difficult to build theoretical deduction in a non-toy environment with thousands of LLM training factors. However, our scaling law can reasonably reflect the learning dynamics of the CPT process, which can be applied in practical CPT scenarios.

## 8. Conclusion

In this study, we explore the learning dynamics in continual pre-training of large language models. We focus on the evolution of performance across general and downstream domains, with domain performance assessed with validation loss. By observations and analyses, we propose a CPT scaling law that integrates distribution shift and learning rate annealing to predict the validation loss at any intermediate training step under common learning rate schedules. Our scaling law provides a comprehensive understanding of key CPT factors and helps optimize the hyper-parameters in CPT for different training goals. Further experiments demonstrate that the law can also be extended to more complicated scenarios such as out-of-domain datasets and pre-trained models with unknown training details. We believe that our CPT scaling law is promising to reshape the understanding of researchers for LLM continual pre-training and scaling laws.

## Impact Statement

CPT is a effective method to enhance the foundation large language models to specific downstream domains or tasks. Our work provides a scaling law to quantitatively describe the learning dynamics of CPT processes. Our results can be used to optimize the training hyper-parameters for balancing the general and downstream performance.

While there will be important impacts resulting from the use of CPT in general, here we focus on the impact of using our scaling law to provide explanations for CPT process. There are many benefits for using our method, such as predicting the loss curve dynamics and optimizing hyper-parameters. The work presented in this aims to advance the field of Large Language Models. There are many potential societal consequences of our work, none which we feel must be specifically highlighted here.

## Acknowledgments

This work was supported by the Strategic Priority Research Program of Chinese Academy of Sciences under Grant XDA0480301 and Fujian Provincial Natural Science Foundation of China (No. 2024J08371).

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

## A. Related Work

**Continual Pre-Training.** Continual Pre-Training (CPT) aims to continuously pre-train LLMs to adapt to new domains, such as code (Hui et al., 2024; DeepSeek-AI et al., 2024), medicine (Chen et al., 2023b) and law (Colombo et al., 2024), while avoiding the need to train domain-specific LLMs from scratch (Shi et al., 2024). The primary objective of CPT is to enhance downstream performance while avoiding catastrophic forgetting (Lange et al., 2023; French, 1999; Gupta et al., 2023; Ibrahim et al., 2024). Most existing CPT methods primarily leverage the replay to mix appropriate pre-training data (Que et al., 2024; Gu et al., 2024) or introduce extra model parameters (Wang et al., 2022) to assimilate new domain knowledge. In this work, we comprehensively study the learning dynamics of CPT and propose a CPT scaling law to describe both general and downstream validation loss.

**Scaling Laws.** Kaplan et al. (2020) empirically discovers a power-law relationship between validation loss $L$ and there factors: model size $N$, dataset size $D$, and training compute. Hoffmann et al. (2022) develops Chinchilla, a compute-optimal LLM to balance model size and dataset size. Tissue et al. (2024) introduces a scaling law to describe the learning dynamics affected by the learning rate annealing, which can predict loss at any training steps under various LRS. However, these scaling laws are limited to pre-training scenarios and do not apply when the training dataset changes.

Regarding continual pre-training, Hernandez et al. (2021b) study scaling law for transfer with respect to model size and CPT data. Barnett (2024) proposes an empirical scaling law that incorporates a transfer gap term to indicate the distribution difference between two datasets. Some methods like D-CPT (Que et al., 2024) and CMR (Gu et al., 2024) introduce scaling laws that account for data replay or mixture ratio in the CPT process. Dou et al. (2024) proposes a quadratic function that considers both learning rate and replay ratio. Nevertheless, these existing scaling laws primarily describe the final loss of given LRS and do not account for all CPT-related factors. Our proposed CPT scaling law integrates all relevant factors and can predict loss at each CPT step, thereby providing a comprehensive description of the complete learning dynamics.

**Hyper-Parameter Optimization.** Identifying optimal hyper-parameter settings is crucial for achieving robust performance in machine learning (Bergstra & Bengio, 2012; Snoek et al., 2012). The principal hyper-parameters in large language models include peak learning rate, learning rate schedules, batch size, and training steps (Kaplan et al., 2020; Hu et al., 2024; Xie et al., 2025). Initial approaches to hyper-parameter optimization primarily utilize model-free techniques such as grid and random search (Bergstra & Bengio, 2012). Subsequently, some methods have employed Bayesian Optimization (Balandat et al., 2020) to predict the performance of various hyper-parameters and select the most effective ones accordingly. Our research focuses on the hyper-parameter in the continual pre-training of larger language models using our proposed CPT scaling law. The hyper-parameter we optimize include the learning rate schedules, peak learning rate, and replay ratio.

## B. Experiment Setups

In this work, we employ multiple experimental setups to validate the effectiveness of our equation. We summarize all experimental setups in Table 1. The majority of our experiments utilize Setting A. Experiments with different replay ratios, batch size, or sequence length are conducted by directly modifying corresponding setups.

## C. Fitting Details

We set $\lambda = 0.999$ in our all experiments. Given the LRS of PT and CPT, we can compute out $S_1^{pt}$, $S_2^{pt}$, $S_1^{cpt}$, and $S_2^{cpt}$ in advance. We adopt a similar fitting method as Chinchilla scaling law (Hoffmann et al., 2022). We minimize the Huber loss (Huber, 1964) between the predicted and the observed log loss using the L-BFGS algorithm (Nocedal, 1980). We implement this by the utilization of `minimize` in `scipy` library. We mitigate the potential issue of local minima of fitting by choosing the optimal fit from a range of initial conditions.

## D. Additional Continual Pre-Training Results

**Prediction of Other LRS.** We use the fitted parameters in Fig. 3 to predict the loss of other LRS (Fig. 10a and Fig. 10d). Our equation could effectively predict the loss of other LRS as shown in Fig. 10.

**Other $D_{cpt}$ Dataset.** Besides Knowledge-Pile (Fei et al., 2024), we also use Eq. 4 to fit transfer loss curves of other $D_{cpt}$ dataset Pile-of-Law (Henderson* et al., 2022) in the Fig. 11.

*Table 1.* Experimental settings adopted in this work. Model size denotes the number of nonembedding parameters. We use AdamW Optimizer (Kingma & Ba, 2015; Loshchilov & Hutter, 2017). Most experiments adopt LLaMA-3's tokenizer (Dubey et al., 2024).

| Setups | Setting A (main) | Setting B | Setting C | Setting D |
|---|---|---|---|---|
| **Model Size** | 106M | 106M | 594M | 1720M |
| **PT dataset** | FineWeb | FineWeb | FineWeb | FineWeb |
| **CPT dataset** | Knowledge-Pile | Pile-of-Law | Knowledge-Pile | Knowledge-Pile |
| **Peak LR** | $2 \times 10^{-4}$ | $2 \times 10^{-4}$ | $2 \times 10^{-4}$ | $2 \times 10^{-4}$ |
| **PT Batch Size (Tokens)** | 4M | 4M | 4M | 4M |
| **CPT Batch Size (Tokens)** | 4M | 4M | 4M | 4M |
| **PT Sequence Length** | 4096 | 4096 | 4096 | 4096 |
| **CPT Sequence Length** | 4096 | 4096 | 4096 | 4096 |
| **Tokenizer** | LLaMA-3's | LLaMA-3's | LLaMA-3's | LLaMA-3's |
| $\beta_1, \beta_2$ **in AdamW** | 0.9, 0.95 | 0.9, 0.95 | 0.9, 0.95 | 0.9, 0.95 |
| **Weight Decay** | 0.1 | 0.1 | 0.1 | 0.1 |
| **Gradient Clip** | 1.0 | 1.0 | 1.0 | 1.0 |

| Setups | LLaMA3.2-1B |
|---|---|
| **Model Size** | 1B |
| **PT dataset** | Unknown |
| **CPT dataset** | Pile-of-Law |
| **Peak LR** | $2 \times 10^{-5}$ |
| **PT Batch Size (Tokens)** | Unknown |
| **CPT Batch Size (Tokens)** | 4M |
| **PT Sequence Length** | Unknown |
| **CPT Sequence Length** | 4096 |
| **Tokenizer** | LLaMA-3's |
| $\beta_1, \beta_2$ **in AdamW** | 0.9, 0.95 |
| **Weight Decay** | 0.1 |
| **Gradient Clip** | 1.0 |

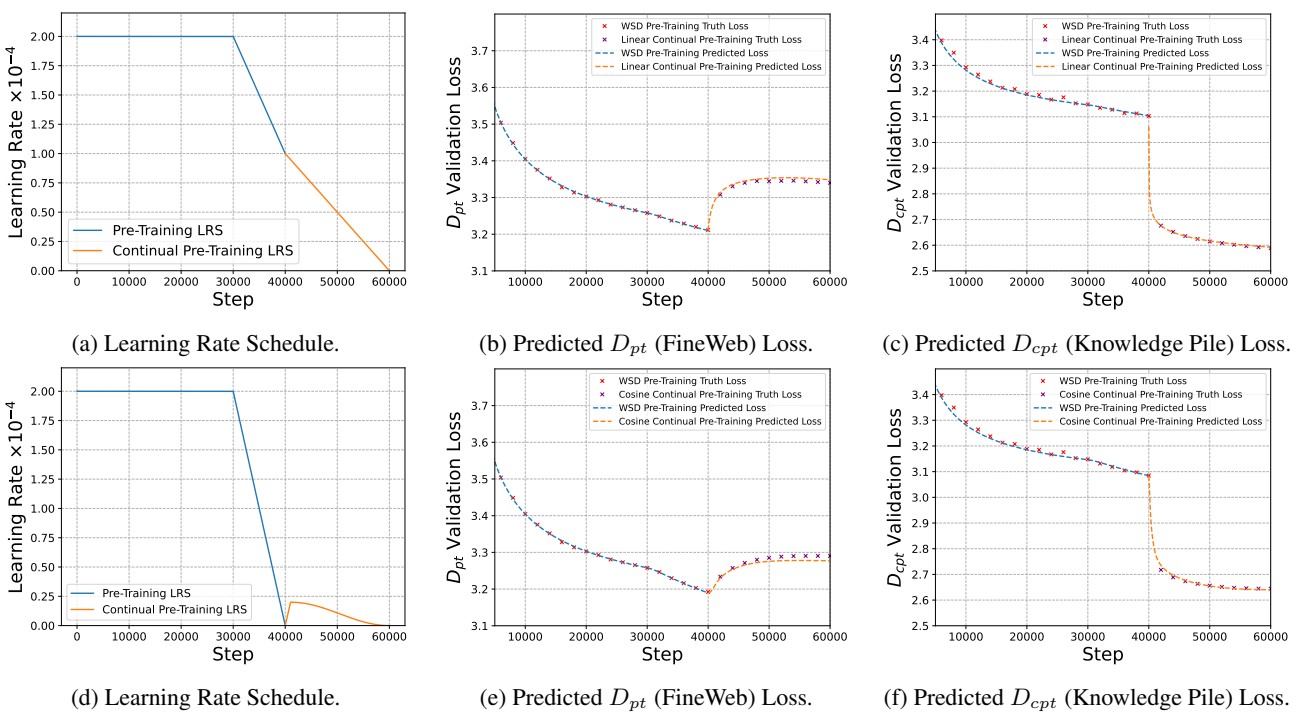

*Figure 10.* Using the fitted parameters in the Fig. 3 to predict all pre-training and CPT loss curve of other LRS. (a) is one kind of without re-warmup method and (b) is a more realistic LRS that the learning rate re-warmup to 10% peak PT learning rate and then annealing to zero with cosine method.

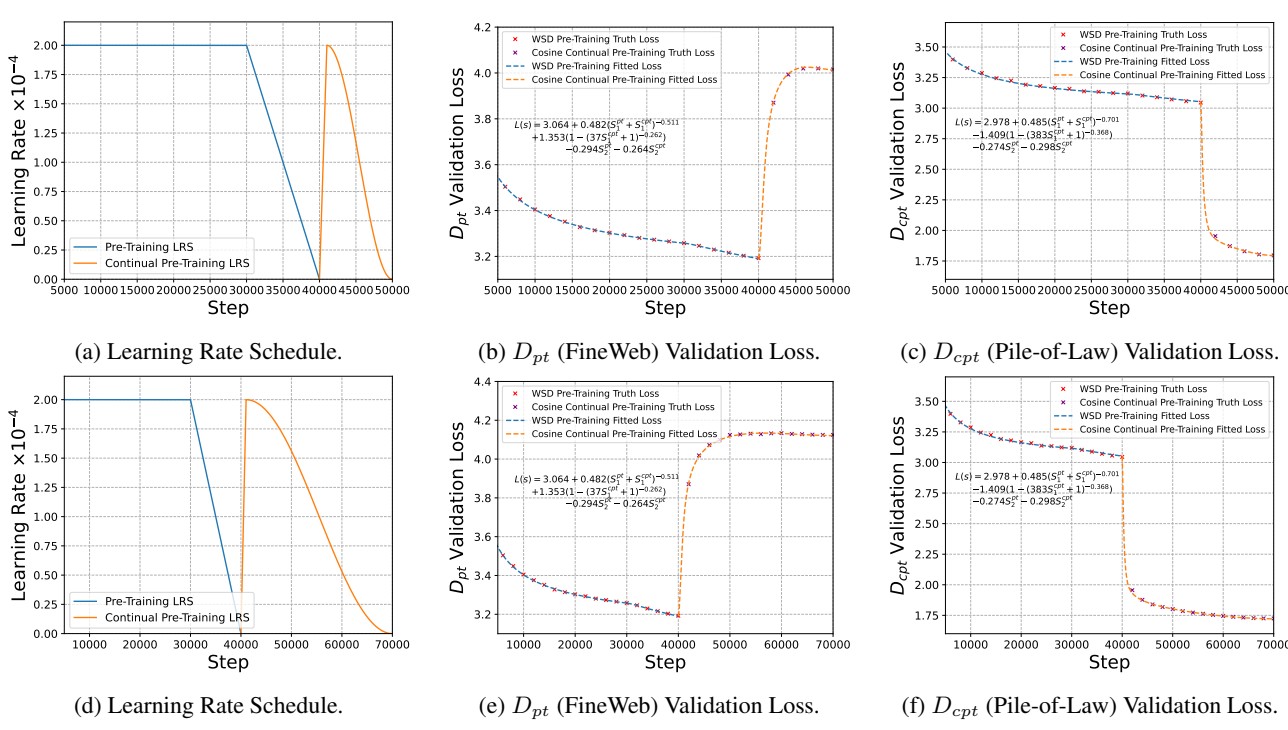

*Figure 11.* Using Eq. 4 to fit all loss curves which are pre-trained with FineWeb and continual pre-trained with law.

(a) Learning Rate Schedule.

(b) $D_{pt}$ (FineWeb) Loss of 33% Replay.

(c) $D_{cpt}$ (KP) Loss of 33% Replay.

(d) Learning Rate Schedule.

(e) $D_{pt}$ (FineWeb) Loss of 50% Replay.

(f) $D_{cpt}$ (KP) Loss of 50% Replay.

(g) Learning Rate Schedule.

(h) $D_{pt}$ (FineWeb) Loss of 67% Replay.

(i) $D_{cpt}$ (KP) Loss of 67% Replay.

*Figure 12.* Using Eq. 4 to fit different $D_{pt}$ replay ratio models independently.

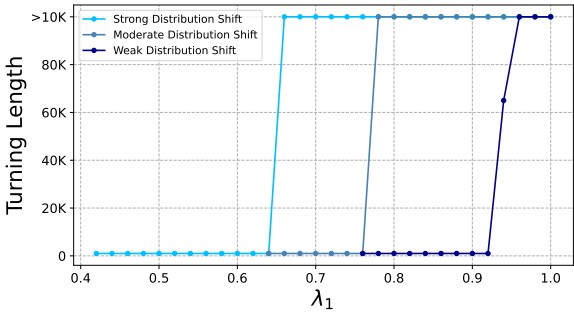

*Figure 13.* The CPT turning lengths of different coefficients for balancing general and downstream domain performance.

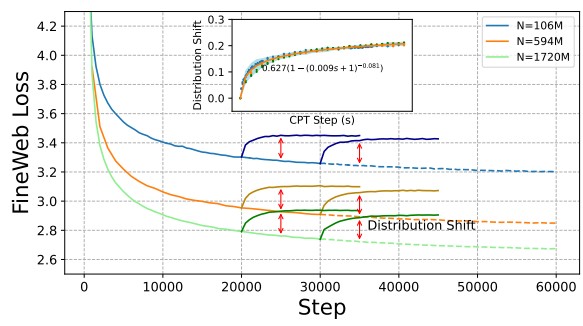

(a) $D_{pt}$ Distribution Shift of different model size $N$.

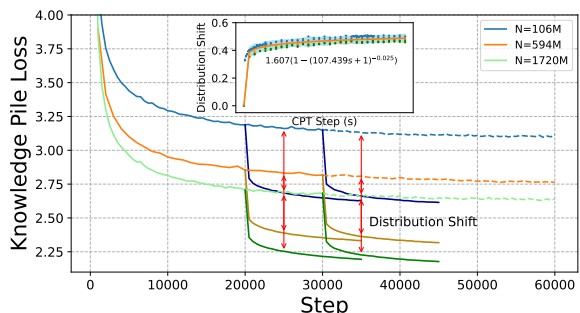

(b) $D_{cpt}$ Distribution Shift of different model size $N$.

*Figure 14.* The distribution shift across different model size with constant LRS.

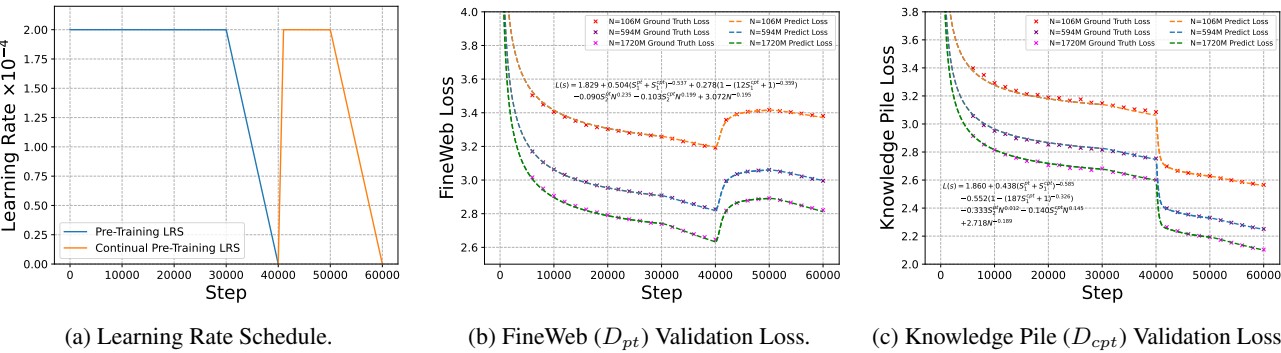

(a) Learning Rate Schedule.      (b) FineWeb ($D_{pt}$) Validation Loss.      (c) Knowledge Pile ($D_{cpt}$) Validation Loss.

*Figure 15.* Using Eq. 7 to fit the loss curve of all model size in both $D_{pt}$ and $D_{cpt}$ validation set.

**Different Replay Ratio.** We use Eq. 4 to fit all loss curves of different $D_{pt}$ replay ratio independently in Fig. 12.

## E. Extension To Model Size Scaling

**Distribution Shift Term of Different Model Sizes.** We first explore the effect of model size $N$ on the distribution shift term. CPT experiments are conducted across various model sizes—106M, 594M, and 1.7B without embedding—using a constant learning rate. As shown in Fig. 14a, the shift terms for different model sizes $N$ nearly coincide. Based on these observations, we hypothesize that the distribution shift term is independent of both model size $N$ and transfer starting points. This implies that data transfer results in a consistent loss difference across different models sizes.

**Model Size Scaling.** Meanwhile, scaling law with LR annealing (Tissue et al., 2024) has demonstrated that the learning rate annealing scales with model sizes $N$, that $S_2 \propto N^\gamma$. Building on the experiments and analysis above, we extend our proposed Eq. 4 to incorporate model size scaling:

$$
\begin{aligned}
L(S^{pt}, S^{cpt}) =& L_0 + A \cdot (S_1^{pt} + S_1^{cpt})^{-\alpha} - C_1 \cdot S_2^{pt} \cdot N^{\gamma_1} - C_2 \cdot S_2^{cpt} \cdot N^{\gamma_2} \\
&+ B \cdot (1 - (1 + E \cdot S_1^{cpt})^{-\beta}) + F \cdot N^{-\gamma_3}
\end{aligned}
\tag{7}
$$

where $F, \gamma_1, \gamma_2, \gamma_3$ is the constant parameters. The $F \cdot N^{-\gamma_3}$ is the model size term in traditional Chinchilla scaling law (Hoffmann et al., 2022). We use Eq. 7 to fit the transfer curves of all model sizes as shown in Fig. 15.

Furthermore, we apply Eq. 4 to fit the CPT loss curves of larger model sizes independently, as illustrated in Fig. 16. This demonstrates the adaptability of our equation across various model sizes.

However, it should be emphasized that the model size in our experiments have not yet reached the scale of mainstream LLMs today, so our experimental conclusions regarding model size are based on the assumptions derived from existing

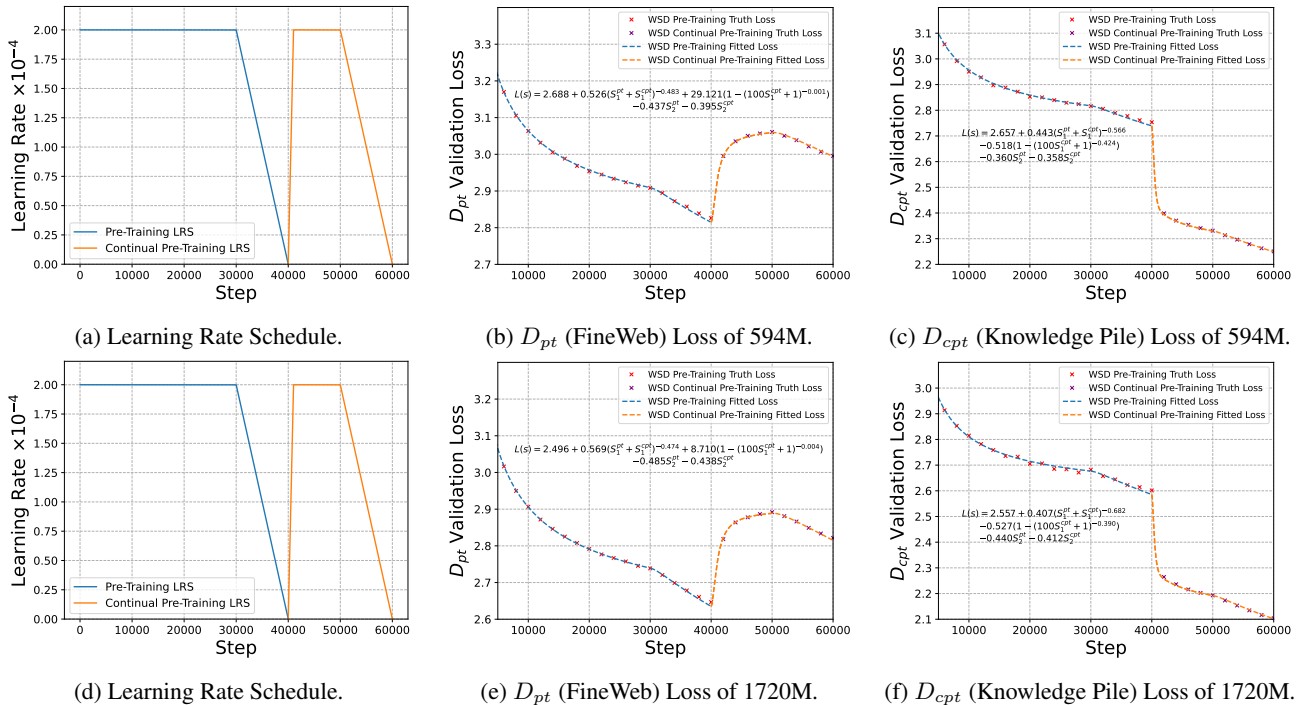

*Figure 16.* Using Eq. 4 to fit all PT and CPT loss curve of 594M and 1720M model size respectively.

results. If the assumption hold for larger model size (e.g., 7B, 70B), we can conclude that *influenced by the same absolute distribution shift value, larger models exhibit greater vulnerability in the general domain but demonstrate better adaptability to downstream domains*.

## F. Batch Size and Sequence Length

In the above experiments, we maintain the same batch size for both PT and CPT phases. However, in the real situation, when computational resources and datasets are limited, practitioners may keep a smaller CPT global batch size than PT phase. Additionally, in other cases, CPT aims to increase the context length of LLMs, requiring increases in both sequence length and RoPE base. We conduct CPT experiments with larger and smaller batch sizes, as shown in Fig. 17. When the sequence length is 8K, we increase RoPE base from 10,000 to 500,000.

**Distribution Shift of Different Batch Size**    We leverage the constant LR to examine whether the distribution shift term for different batch sizes satisfies the same functional form. We using Eq. 4 to fit the loss curves of different batch sizes. As shown in Fig. 17a and Fig. 17b, all loss curves with different transfer steps for both larger and smaller batch sizes can be fitted with a single distribution shift term, which demonstrates that Eq. 4 can also accommodate changes in batch size and sequence length.

## G. Open-Source Pre-Training Models

A more realistic scenario posits that the PT model is an open-source model, and we do not have access to the exact PT process. Therefore, the distribution of PT dataset, the loss potential, and the PT training amount usually remain unknown.

**Unknown PT Training Amount and Loss Potential**    For the open-source models, we do not know the PT training amount and loss potential to get the PT forward area $S_1^{pt}$ and the final LR to calculate the CPT annealing area $S_2^{cpt}$. For forward area $S_1^{pt}$, we treat it as a parameter to be fitted. Typically, most open source models will anneal the LR to zero or a minium LR to get a better benchmark performance. We assume that the final LR of all open-source models is zero, which facilitates the computation of the CPT annealing area $S_2^{cpt}$. The learning rate of CPT of open-source models is consider to re-warmup

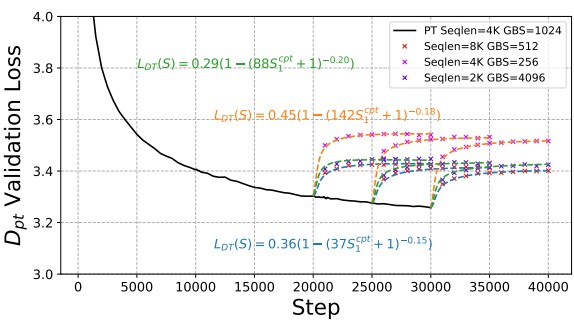 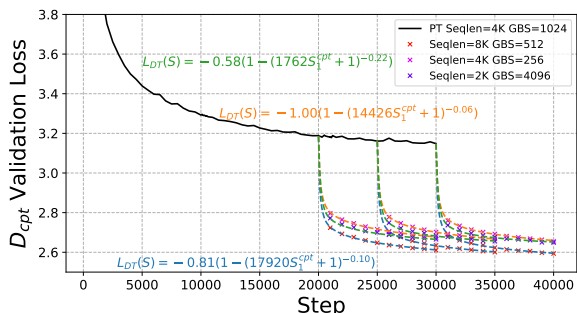

(a) $D_{pt}$ (FineWeb) loss curve of different batch size.  (b) $D_{cpt}$ (Knowledge Pile) loss curve of different batch size.

*Figure 17.* Using Eq. 4 to fitted loss curve of different batch size in the continual pre-traininig. The smaller batch size is 1M tokens with 4K sequence length and the larger batch size is 8M tokens with 8K sequence length. We annotate the different distribution shift terms in the figure.

from zero to the specific peak learning rate and then anneal with specific LRS.

**Unknown PT Dataset Distribution**   The aforementioned equation holds only in the loss curve of $D_{pt}$ and $D_{cpt}$ validation dataset. However, we do not know the exact $D_{pt}$ dataset distribution of open-source models. In this case, we could select an open-source common crawl validation set as **proxy** $D_{pt}$.

To verify the two hypotheses mentioned above are reasonable, We conduct the experiments that continual pre-train the LLaMA3.2-1B (Dubey et al., 2024) with Pile-of-Law dataset (Henderson* et al., 2022). We consider the C4 portion of the RedPajama (Weber et al., 2024) dataset as a **proxy** $D_{pt}$. We use fewer training steps to fit the parameters and then predict the loss of longer steps for the proxy $D_{pt}$ and true $D_{cpt}$. As shown in Fig. 18, Eq. 4 could effectively predict the further loss of **proxy** $D_{pt}$ and $D_{cpt}$. Based on the proxy $D_{pt}$ and true $D_{cpt}$, we could describe the performance dynamics and complete the above hyper-parameters optimization for open-source models.

We also conduct experiments with our model pre-trained with FineWeb and continual pre-trained with Pile-of-Law dataset. We treat it as a model with unknown PT information. We still use the portion of RedPajama as the proxy $D_{pt}$ dataset to predict the loss of longer training steps as shown in Fig. 18.

## H. Adding Replay Ratio to Our Formulation

D-CPT law (Que et al., 2024) proposed a scaling law integrating with $D_{pt}$ and $D_{cpt}$ data mixture ratio. We have also integrated this data mixture ratio into our formulation. Appendix D demonstrates that loss curves for different data ratios can be individually fitted using distinct equations. However, we are currently exploring a unified formulation that incorporates the data mixture ratio to represent all loss curves. Both the distribution shift term and the LR annealing term are influenced by the replay ratio. A higher $D_{pt}$ ratio leads to a weaker distribution shift, and results in a smaller LR annealing term in the $D_{cpt}$ validation loss, while increasing the LR annealing term in the $D_{pt}$ validation loss. We find that the exponential form, which is consistent with the Data Mixing Law (Ye et al., 2024), best fits these effects and subsequently incorporate it into both the distribution shift term and LR annealing term:

$$L_{pt} = L_0 + A \cdot \left(S_1^{pt} + S_1^{cpt}\right)^{-\alpha} - C_1 \cdot S_2^{pt} - C_2 \cdot S_2^{cpt} e^{a_1 r_{pt}} + B \cdot \left(1 - \left(1 + E \cdot S_1^{cpt}\right)^{-\beta}\right)\left(1 - e^{-a_2 r_{cpt}}\right)$$
$$L_{cpt} = L_0 + A \cdot \left(S_1^{pt} + S_1^{cpt}\right)^{-\alpha} - C_1 \cdot S_2^{pt} - C_2 \cdot S_2^{cpt} e^{a_1 r_{cpt}} + B \cdot \left(1 - \left(1 + E \cdot S_1^{cpt}\right)^{-\beta}\right)\left(e^{a_2 r_{cpt}} - 1\right)$$
(8)

where $r_{pt}$ and $r_{cpt}$ are the data mixture ratio of PT and CPT data respectively, such that $r_{pt} + r_{cpt} = 1$, and $a_1$ and $a_2$ are the additional parameters. To ensure that the distribution shift term is zero when $r_{cpt}$ equals zero, we have modified the exponential formulation in the distribution shift term accordingly. The effectiveness of this equation is illustrated in Fig. 19. While the D-CPT law predicts only the final loss across different replay ratios, our method is capable of describing the entire training dynamics for various replay ratios.

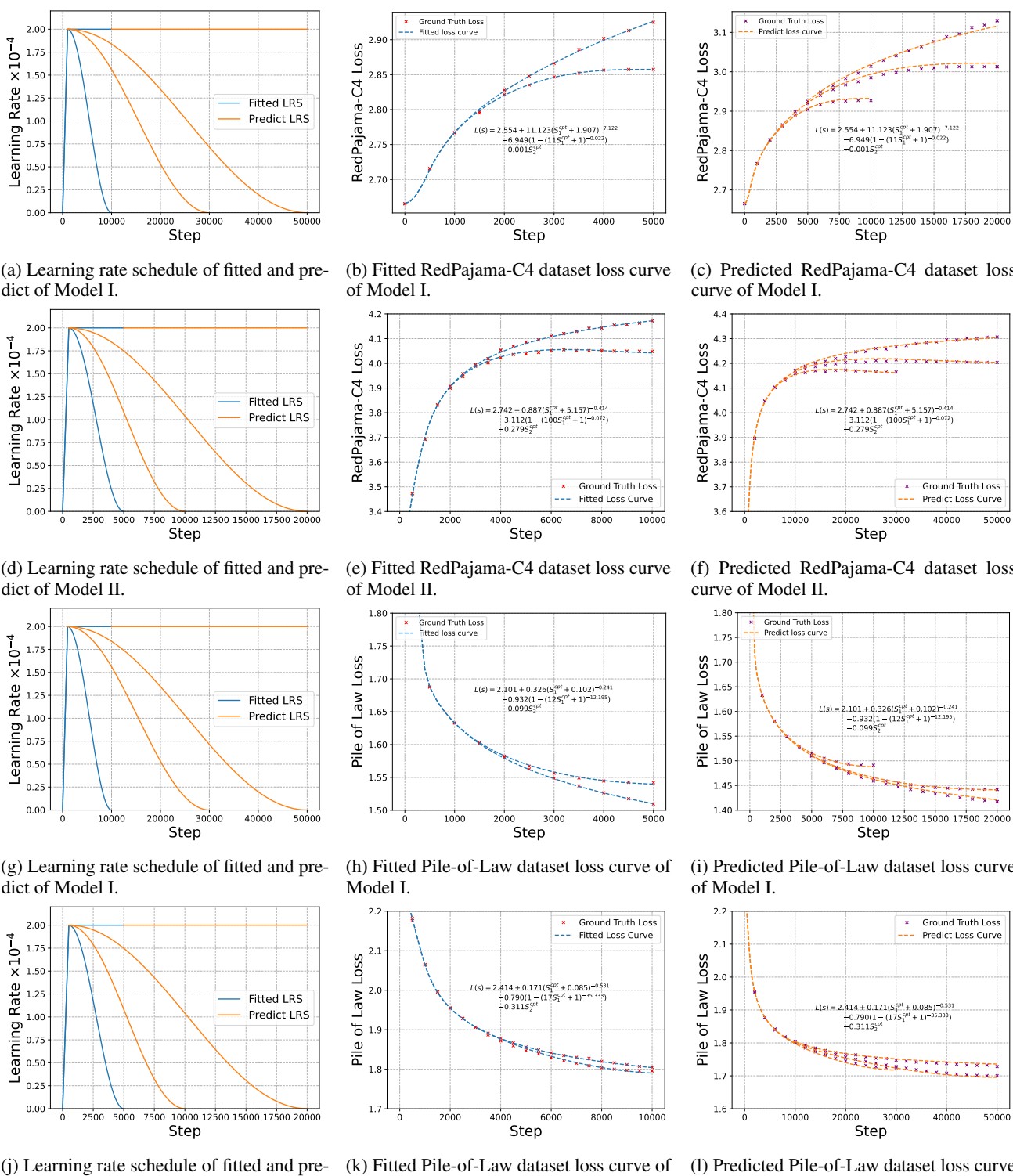

(a) Learning rate schedule of fitted and predict of Model I.

(b) Fitted RedPajama-C4 dataset loss curve of Model I.

(c) Predicted RedPajama-C4 dataset loss curve of Model I.

(d) Learning rate schedule of fitted and predict of Model II.

(e) Fitted RedPajama-C4 dataset loss curve of Model II.

(f) Predicted RedPajama-C4 dataset loss curve of Model II.

(g) Learning rate schedule of fitted and predict of Model I.

(h) Fitted Pile-of-Law dataset loss curve of Model I.

(i) Predicted Pile-of-Law dataset loss curve of Model I.

(j) Learning rate schedule of fitted and predict of Model II.

(k) Fitted Pile-of-Law dataset loss curve of Model II.

(l) Predicted Pile-of-Law dataset loss curve of Model II.

*Figure 18.* Using Eq. 4 to fit and predict the proxy $D_{pt}$ and true $D_{cpt}$ dataset of open-source PT models. The Model I refers to LLaMA3.2-1B. Model II refers to our model pre-trained with FineWeb but we regard it as an unknown model and use proxy $D_{pt}$ rather than FineWeb. The $D_{cpt}$ dataset are both Pile-of-Law, and the proxy $D_{pt}$ is RedPajama-C4.

## I. Optimal Hyper-Parameters

In the Fig. 8, we show the optimal hyper-parameters based on different coefficients. For optimal loss potential and peak LR, we use FineWeb as $D_{pt}$ and and explore three different $D_{cpt}$ dataset: (1) Pile of Law, (2) Knowledge Pile, and (3) a mixture of 67% FineWeb and 33% Knowledge Pile. These three $D_{cpt}$ datasets have strong, moderate, and weak distribution shifts comparing to $D_{pt}$. The training setting for each $D_{cpt}$ dataset is consistent with the Setting A and Setting B in Table 1. The LRS used for fitting these three $D_{cpt}$ datastes and the fitted equation coefficients are shown in Fig. 11, Fig. 3 and Fig. 12. We directly use these three sets of coefficients to search the optimal hyper-parameters. In the Fig. 8a and Fig. 8b, we assume that the LRS for the PT phase follows a WSD schedule with 40k steps, while the continual CPT phase employs a cosine schedule with 10k steps, consistent with the configuration shown in Fig. 11a.

For optimal replay ratio, we use FineWeb as $D_{pt}$ and Knowledge Pile as $D_{cpt}$. In Fig. 8c, we maintain the assumption that the PT phase employs WSD scheduling while the CPT phase uses cosine scheduling with varying CPT step counts. The blue dashed reference line represents the scenario where the target weight ($\lambda_1$) equals the replay ratio. It can be reasonably assumed that if the model were initialized from scratch (rather than from a pre-trained model), the optimal replay ratio curve would follow the blue dashed line. However, since our model is pre-trained, this causes the curve to deviate and exhibit a wavy pattern. We directly apply the fitted coefficients presented in Fig. 19 to determine the optimal replay ratio.

## J. Other Formats with LR Annealing

Similar with scaling law with LR annealing (Tissue et al., 2024), we also try the other possible forms of LR annealing.

**Adding a LR-weighted Coefficient**    To solve that when LR anneals to nearly 0, $S_2$ still has historical momentum, making the loss continue to decrease. A revision is that adding a LR-weighted coefficient to $S_2$:

$$S_2 = \sum_{i=1}^{s} m_i \cdot \eta_i^{\epsilon} \tag{9}$$

We test the coefficient $\epsilon$ is 0.1 and 0.2, showing the fitted result in the Fig. 20.

$S_2$ **Power Formats**    Considering that the annealing loss and $S_2$ have a positive correlation, $L \propto S_2^{\zeta}$ might be a more reasonable format than $L \propto S_2$. We revise our formulation:

$$\begin{aligned} L = L_0 &+ A \cdot \left( S_1^{pt} + S_1^{cpt} \right)^{-\alpha} - C_1 \cdot (S_2^{pt})^{\zeta_1} - C_2 \cdot (S_2^{cpt})^{\zeta_2} \\ &+ B \cdot \left( 1 - \left( 1 + E \cdot S_1^{cpt} \right)^{-\beta} \right) \end{aligned} \tag{10}$$

We add two other fitted parameters in the function for different annealing area of PT and CPT. We also show the fitted effect in the Fig. 20. We show the huber loss and $R^2$ of all possible formats in the Table 2. All the fitting effect are really good, but the original format has the fewest parameters which is more effective.

## K. Out-of-Domain Validation Set

Data mixing law (Ye et al., 2024) shows that validation loss for some domains can be represented by a combination of other domains. In scenarios where the CPT dataset, such as Knowledge-Pile, is not highly domain-specific, employing a linear combination of $D_{pt}$ and $D_{cpt}$ can serve as a reasonable approximation for certain downstream validation sets. However, it is important to note that this approach may not be universally applicable across all CPT datasets and all $D_{ood}$ validation loss scenarios. We test the validity of using a linear combination of $D_{pt}$ (FineWeb) and $D_{cpt}$ (Knowledge-Pile) to estimate the validation loss for certain out-of-domain sets in the Fig. 21 and Fig. 22. These out-of-domain validation sets include StackExchange, arXiv, and C4 in RedPajama (Weber et al., 2024), as well as PhilPapers and Books in Pile (Gao et al., 2020), SlimPajama (Soboleva et al., 2023), and Open-Web-Math (Paster et al., 2023).

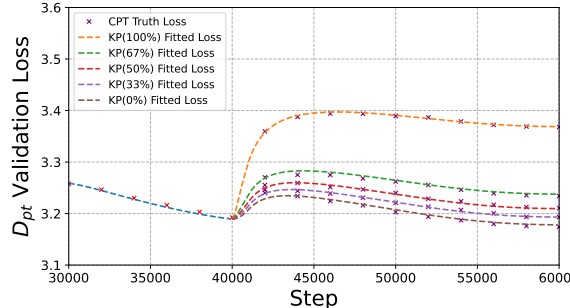

(a) $D_{pt}$ loss curve of different replay ratios.

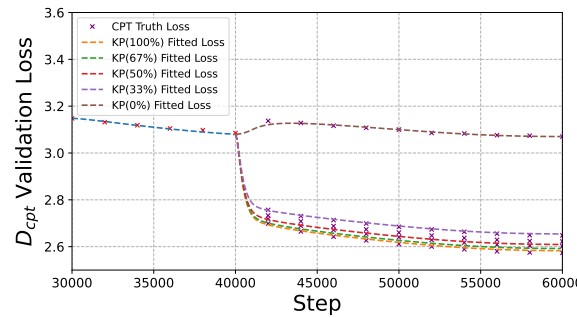

(b) $D_{cpt}$ loss curve of different replay ratios.

*Figure 19.* Using Eq. 8 to fitted all loss curves of different replay ratio in the continual pre-training. The fitted equation is $L_{pt} = 3.067 + 0.480 \cdot \left(S_1^{pt} + S_1^{cpt}\right)^{-0.510} - 0.280 \cdot S_2^{pt} - 0.263 \cdot S_2^{cpt} e^{0.055 r_{pt}} + 0.276 \cdot \left(1 - \left(1 + 99.35 \cdot S_1^{cpt}\right)^{-\beta}\right) \left(1 - e^{-3.238 r_{cpt}}\right)$ and $L_{cpt} = 2.992 + 0.456 \cdot \left(S_1^{pt} + S_1^{cpt}\right)^{-0.510} - 0.285 \cdot S_2^{pt} - 0.279 \cdot S_2^{cpt} e^{0.037 r_{pt}} - 0.526 \cdot \left(1 - \left(1 + 100.34 \cdot S_1^{cpt}\right)^{-\beta}\right) \left(e^{5.696 r_{cpt}} - 1\right)$.

*Table 2.* The fitting effect of different possible equation formats.

| $D_{pt}$ | Huber Loss $\downarrow$ | $R^2 \uparrow$ |
|---|---|---|
| Original | 0.0016 | 0.9944 |
| Adding LR Cofficient ($\epsilon = 0.1$) | 0.0016 | 0.9950 |
| Adding LR Cofficient ($\epsilon = 0.2$) | 0.0017 | 0.9950 |
| Adding $S_2$ Power | 0.0016 | 0.9952 |

| $D_{cpt}$ | Huber Loss $\downarrow$ | $R^2 \uparrow$ |
|---|---|---|
| Original | 0.0021 | 0.9993 |
| Adding LR Cofficient ($\epsilon = 0.1$) | 0.0025 | 0.9984 |
| Adding LR Cofficient ($\epsilon = 0.2$) | 0.0024 | 0.9983 |
| Adding $S_2$ Power | 0.0025 | 0.9984 |

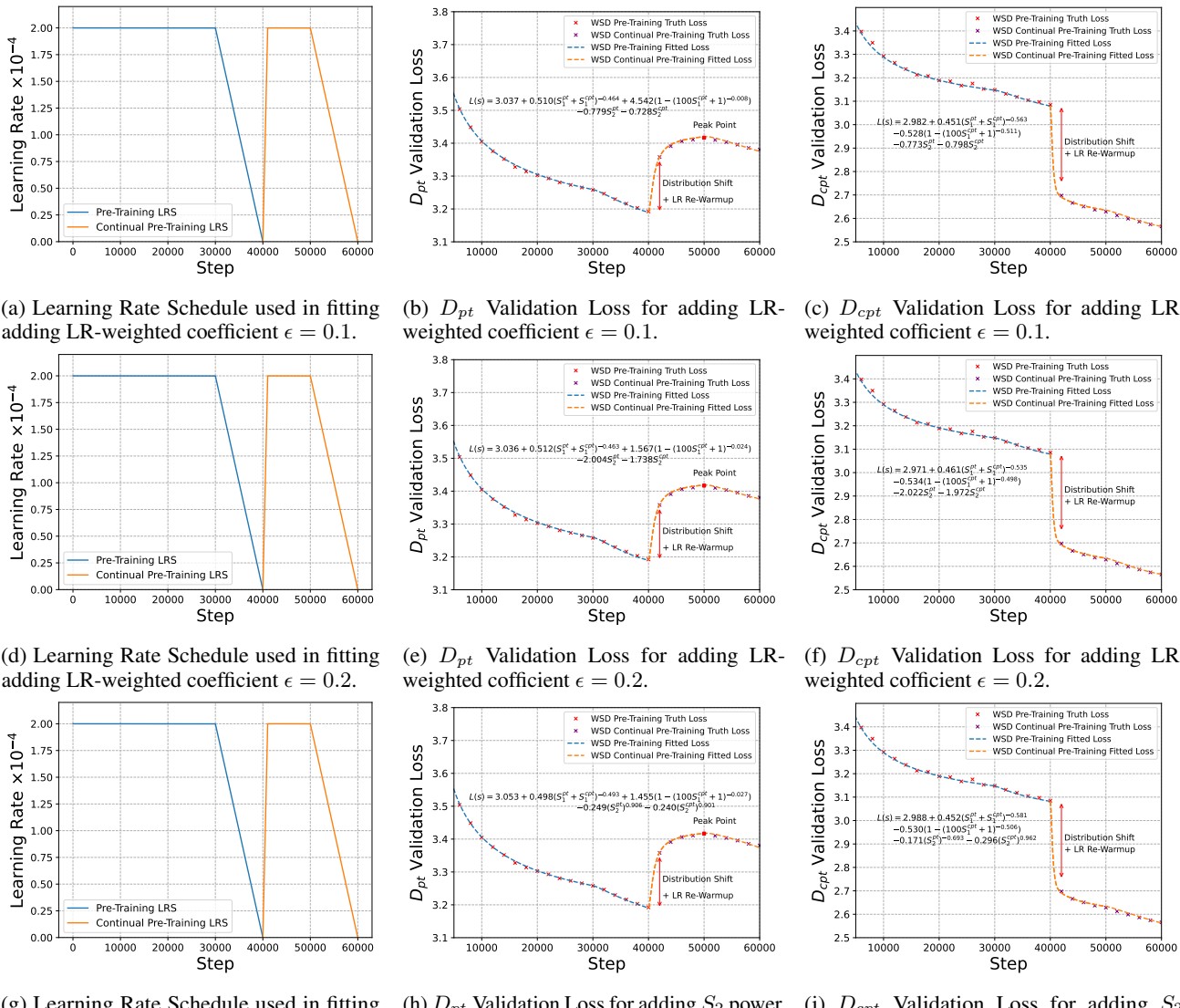

(a) Learning Rate Schedule used in fitting adding LR-weighted coefficient $\epsilon = 0.1$.

(b) $D_{pt}$ Validation Loss for adding LR-weighted coefficient $\epsilon = 0.1$.

(c) $D_{cpt}$ Validation Loss for adding LR-weighted cofficient $\epsilon = 0.1$.

(d) Learning Rate Schedule used in fitting adding LR-weighted coefficient $\epsilon = 0.2$.

(e) $D_{pt}$ Validation Loss for adding LR-weighted cofficient $\epsilon = 0.2$.

(f) $D_{cpt}$ Validation Loss for adding LR-weighted cofficient $\epsilon = 0.2$.

(g) Learning Rate Schedule used in fitting adding adding $S_2$ power format.

(h) $D_{pt}$ Validation Loss for adding $S_2$ power format.

(i) $D_{cpt}$ Validation Loss for adding $S_2$ power format.

*Figure 20.* Using other possible $S_2$ formats Eq. 4 to fit all PT and CPT loss curve with different LRS.

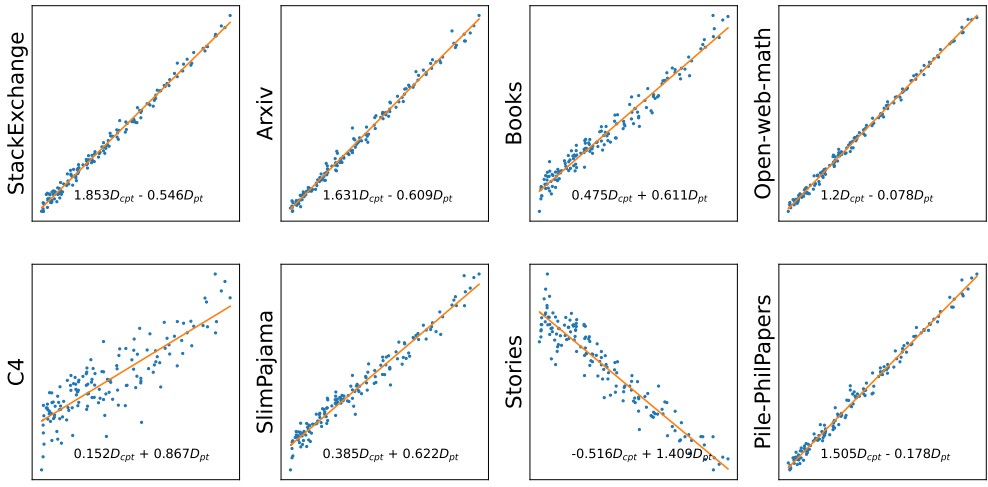

*Figure 21.* The linear combination of $D_{pt}$ and $D_{cpt}$ to represent the out-of-doamin $D_{ood}$ validation loss.

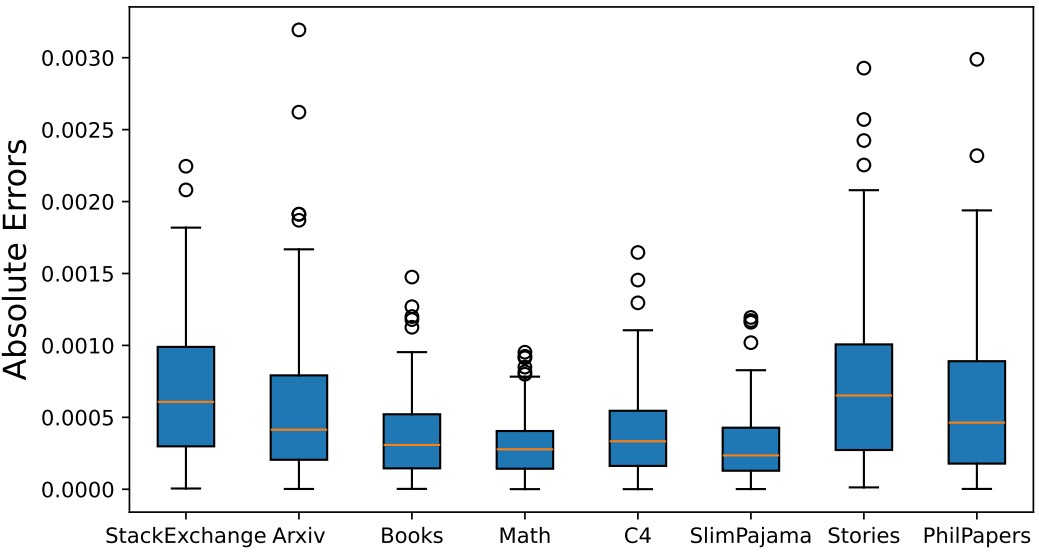

*Figure 22.* The absolute errors of linear combination of $D_{pt}$ and $D_{cpt}$ to represent the out-of-doamin $D_{ood}$ validation loss.

