# OpenReview forum: "Learning Dynamics in Continual Pre-Training for Large Language Models"
_ICML.cc/2025/Conference — ICML 2025 oral_

### Official Review · Reviewer_oHxM · 2025-03-14

**Overall Recommendation:** 3

**Summary:**

This paper studies the training dynamics during continual pre-training. Specifically, they study the scaling law of the loss curve during continual pre-training (CPT), which considers the scaling law due to learning rate annealing and the scaling law due to the distribution shift. Based on this formulation, they consider several factors in CPT, including loss potential, distribution shift between pre-training and CPT dataset, peak learning rate, and CPT steps. They also discuss how the proposed scaling law can be accurately fitted for open-source models.

**Claims And Evidence:**

I do not find all the claims to be supported by the evidence. However, I believe this is somewhat related to the presentation issue of the paper, making it hard for me to know what the paper is saying. For example, on page 3, the paper writes, " As shown in Fig. 2, these distribution shift terms tend to overlap at each transfer starting point." I am not sure where the overlap referred to here is. In this case, it would be hard to evaluate how well-supported this claim is.

**Essential References Not Discussed:**

N/A

**Experimental Designs Or Analyses:**

The experiments use multiple models of the same architecture with different sizes. The issue with the evaluation is mostly related to the choice of the CPT dataset, which I elaborated previously.

**Methods And Evaluation Criteria:**

This paper uses FineWeb for pre-training and Knowledge-Pile for CPT. Using FineWeb as a pre-training dataset is reasonable. However, only using Knowledge-Pile as the CPT data is a weakness of this paper. CPT can be conducted on coding or mathematical datasets to improve the specific ability. It is unclear whether the observation and conclusion in this paper hold for CPT datasets in other domains or different datasets. Using Knowledge-Pile may also be questionable since this is not a dataset that the research community has a consensus on its quality, at least judging by the review from ICLR 2025.

**Other Comments Or Suggestions:**

- Is the caption in Figue 5 (b) (c) (e) (f) wrong?

**Other Strengths And Weaknesses:**

Strengths
==
I found this paper to be quite interesting, and it provided many useful and intriguing observations. The findings highlighted in this paper should be of interest for researchers in the community

Weakness
==
The key weakness of this paper, and the reason I am currently leaning toward rejecting it, is the presentation issue, which makes the paper difficult to read.
 - Acronyms are not properly explained. For example, the term "LRS" is never explained. The term "WSD" is not clearly explained when it first appeared in the paper.
  - The term "transfer curve" is not explained. This is not a formal term in math, so I am not sure what the author is referring to when using this term.
   - The explanation of Figure 1 is not sufficient when it is presented. In Line 107 and the "Observation" in Section 2.1, the paper refers to Figure 1 without explaining many details, including the LRS, WSD, hidden pre-training, and transfer point. It is not easy for the reader to understand this figure.
   - The font size in many figures is too small to read. The legends in Figures 1, 2, and 3 are completely unreadible when printed on an A4 paper.
   - The term "loss potential" appears in the abstract and introduction but is explained in Section 3.3

**Questions For Authors:**

- In Figure 1, is the hidden pre-training the second epoch of the same pre-training data?

**Relation To Broader Scientific Literature:**

This paper proposes a scaling law for CPT, which has not been discussed in prior literature. They also discuss how the proposed scaling law can help understand the contribution of different factors in CPT and how to choose the best hyperparameters. These contributions are unique and not seen in prior works

**Theoretical Claims:**

There are no theoretical claims.

---

> ### Author Rebuttal · Authors · 2025-03-31
>
> We sincerely appreciate your thoughtful suggestions and valuable feedback.
>
> **Response to "Claims and Evidence about sentences that are hard to understand":**
> We apologize for misunderstanding caused by our use of the term "overlap." Our intended meaning was that the curves from different transfer starting points "coincide". In the subgraph of Figure 2, we show the distribution shift curves of different transfer starting points and the horizontal axis represents the number of steps in CPT. These distribution shift curves almost "coincide" and could be fitted with a power-law form.
>
> **Response to "Choosing Knowledge-Pile":**
> First, we do not exclusively utilize Knowledge-Pile as the CPT dataset. We also employ Pile-of-Law [1] as a domain-specific CPT dataset and demonstrate the fitting and predictive capabilities of our scaling law in Appendix D. Second, the data in Knowledge-Pile [2] is derived and filtered from existing data sources, such as Wikipedia, arXiv, Semantic Scholar, without any atypical components. From the components [2], Knowledge-Pile is a good combination with higher STEM knowledge density, which can be used to enhance domain-specific abilities. Third, Our law is independent with the CPT dataset. When the CPT dataset distribution or quality change (better or worse), the form of our law remains applicable while the coefficients differ.
>
> **Response to "Weakness":**
> **W1: "LRS" and "WSD"**:
> We apologize for the reduced readability caused by the abbreviation of certain concepts in Figure 1. LRS refers to learning rate schedules, and WSD [3] denotes the Warmup-Stable-Decay schedule. We will implement these clarifications in the final version.
> **W2: Explain "transfer curve"**:
> The term "transfer curve" is actually the CPT curve. As stated in line 96, "the CPT loss curve is a transitional curve on both D_pt and D_cpt validation set." The CPT curves in Figure 1 (b), (c), (e), and (f) illustrate that the curves deviate from the blue dashed line, and finally align with the orange dashed line.
> **W3: Explain "transfer point" and "Hidden Pre-training"**:
> The "transfer starting point" represents the starting step point of CPT. We formally define the "Hidden Pre-training" curve in Section 2.2 and represent these trajectories with dashed lines in Figure 1.
> We acknowledge that the definations of some terms were not placed appropriately, and we commit to refining these definitions in the final version.
> **W4: Small font size**:
> We apologize for the small font size in the figures and we will increase the font size in the final version.
> **W5:  Explain "loss potential"**:
> In the final version of the manuscript, we will incorporate a comprehensive description of the "loss potential" concept within the introduction section to establish a clearer theoretical foundation from the outset.
>
> **Response to "Comments and suggestions: The wrong caption in Figue 5 (b)(c)(e)(f)?":**
> We sincerely apologize for unclear captions of Figure 5 due to our intention to save space during the writing process.
> The complete caption in the Figure 5 is:
> (b) D_cpt true loss v.s. CPT step of different loss potentials (w/o re-warmup setting)
> (c) D_cpt predicted loss v.s. loss potentials of different CPT steps (w/o re-warmup setting)
> (e) D_cpt true loss v.s. CPT step of different loss potentials (w/ re-warmup setting)
> (f) D_cpt predicted loss v.s. loss potentials of different CPT steps (w/ re-warmup setting)
>
>
> **Response to "Questions: is the hidden pre-training the second epoch of the same pre-training data?":**
> No. The hidden pre-training is not from the second epoch. it is based on the different data from the same distribution. For example, in our experiment, we continually pre-train the model on the remaining PT dataset Fine-web.
>
>
> [1] Henderson, Peter, et al. "Pile of law: Learning responsible data filtering from the law and a 256gb open-source legal dataset." Advances in Neural Information Processing Systems 35 (2022): 29217-29234.
> [2] Fei, Zhaoye, et al. "Query of cc: unearthing large scale domain-specific knowledge from public corpora." arXiv preprint arXiv:2401.14624 (2024).
> [3] Hu, Shengding, et al. "Minicpm: Unveiling the potential of small language models with scalable training strategies." arXiv preprint arXiv:2404.06395 (2024).

---

> > ### Comment · Reviewer_oHxM · 2025-04-02
> >
> > Thank you for your explanation. I trust that the author will resolve the clarity issue in future revisions. I believe this can be easily resolved, so I raise my score.

---

### Official Review · Reviewer_5BNQ · 2025-03-17

**Overall Recommendation:** 4

**Summary:**

This paper explores the learning dynamics of continual pre-training and proposes scaling laws for the same. Based on the proposed scaling laws, the authors discuss several critical factors in continual pre-training. Overall, this is an educational paper to understand how continual pre-training works.

**Claims And Evidence:**

Yes.

**Essential References Not Discussed:**

Nothing I am aware of.

**Experimental Designs Or Analyses:**

Yes. The experimental designs in this paper are well thought out. There is a lot of thorough analysis.

**Methods And Evaluation Criteria:**

Yes.

**Other Comments Or Suggestions:**

1. "loss potential" and "turning length" are used in the introduction without proper definition of them. So it is hard to understand the last paragraph of section 1.

2. In page 7, for Fig 13, it is worth mentioning that the figure is in the appendix.

**Other Strengths And Weaknesses:**

This paper explores the effect of several hyperparameters including learning rate, replay ratio during continual pre-training. Overall, this is a great educational paper to learn more about the learning dynamics of CPT.

**Questions For Authors:**

1. In page 5, you mention "models with higher loss potential achieve lower final losses". Why is that the case?

2. Finding-3 is not clear. What do you mean by releasing the high loss potential version?

3. Finding-4 is also not clear to me. Can you please explain it?

4. How do you set lambda_1 and lambda_2 in equation 5? You mention that it is based on practical requirements. But it is not clear what these requirements could be.

**Relation To Broader Scientific Literature:**

The authors have done a good job of relating the work to the existing works.

**Theoretical Claims:**

There are no proofs in the paper. There are several claims about the scaling laws which are derived from other works. I did not check the correctness of the scaling laws since I am not an expert in scaling laws. I will let other reviewers judge this aspect.

---

> ### Author Rebuttal · Authors · 2025-03-31
>
> We sincerely appreciate your thoughtful suggestions and valuable feedback.
>
> **Response to "Comments Or Suggestions 1 about loss potential and turning length":**
> We apologize for any poor reading experience caused by organizational issues about these terms in our paper and will address these concerns in the final version.
> For loss potential, we define in line 187 "captures the potential of future loss drop via LR annealing.", which corresponds to the height in Figure 4(c). For example, if the pre-training implements substantial annealing and end at the valley bottom in Figure 4(c), the remaining annealing room or potential for CPT is diminished. Quantitatively, we define loss potential as the ratio of the final learning rate of the pre-training annealing phase to the initial or maximum learning rate in the pre-training phase: $ loss\ potential = \frac{Final\ LR\ of\ PT}{Maximum\ LR\ of\ PT} $, which is same as the legend in our Figure 5(b)(e) and x-axis of Figure 5(c)(f). In Figure 5(a)(d), if the pre-training model anneals from 2e-4 to 1e-4 using a linear method, the loss potential for this model is 50%.
> For turning length, we define it in line 315 as "The minimum training steps required to achieve a lower loss are designated as the turning length." We also illustrate the meaning of this term in Figure 7(c). Qualitatively, when the CPT steps once reach the turning length, the validation loss on pre-training dataset reverts to the same loss as the very beginning of CPT as shown in Figure 7(c).
>
> **Response to "Comments Or Suggestions 2 about Figure 13":**
> We will implement appropriate revisions for Figure 13 in the final version of the paper.
>
> **Response to "Q1: Why is models with higher loss potential achieve lower final losses":**
> The statement "models with higher loss potential achieve lower final losses" indicates that pre-training models with higher loss potential can attain lower CPT validation losses and demonstrate superior adaptation to the CPT dataset.
> The intuitive understanding is that models with higher loss potential can allocate more annealing to the CPT phase, which enable to achieve lower D_cpt validation loss. The more specific reason of this finding is explained in line 255-265 in our paper. This finding is substantiated by the truth training loss trajectories of different loss potential models, as illustrated in Figure 5(b) and 5(e), as well as by the predicted loss values derived from our CPT law, as depicted in Figure 5(c) and 5(f).
>
> **Response to "Q2: What do you mean by releasing the high loss potential version":**
> Based on the above explanation, the models with higher loss potential could better adapt to the CPT dataset. However, current open-source models, such as Qwen or Llama, typically employ extensive learning rate annealing that reduces the learning rate to near-zero or minimal values, resulting in low loss potential. If releasing high loss potential variants of these models specifically (i.e., those trained without learning rate annealing), researchers and practitioners could adapt these models for CPT or downstream tasks.
>
> **Response to "Q3: Explain Finding 4":**
> The turning length is affected by: (1) the distribution distance between PT and CPT data; and (2) the sufficiency of pretraining. If the distribution distance is large or the pre-training is sufficient, the turning length becomes larger and even to potentially infinite. The detailed explanation is in the line 308-315 and Figure 7(c). If you have further questions about the Finding 4, feel free to further comment.
>
> **Response to "Q4: How do you set lambda_1 and lambda_2 in equation 5?":**
> It can be considered in two scenarios.
> In some situations, we could allocate a percentage ratio for validation losses on PT and CPT datasets based on our prior knowledge with the importance between general and downstream performance.
> Otherwise, for a specific test set that we focus and optimize, we could always precompute the linear coefficients based on dataset similarity, as demonstrated in Section 5.2.

---

### Official Review · Reviewer_oDDW · 2025-03-18

**Overall Recommendation:** 3

**Summary:**

This paper performs an empirical study of loss curves during continual pre-training. The paper seeks to characterize how the training loss on a new dataset will evolve when a pretrained model is subjected to new data in a CPT setup. To that end, the authors present a series of experimental setups showing how loss behaves in different scenarios. CPT loss curve is shown to be decomposable into previously characterized scaling law with learning rate annealing and a novel power-law term for capturing distribution shift between pretraining and continued pre-training data domains. The authors show in many setups that the laws proposed fit nicely the empirical data. The authors discuss the formulation and fitting of the laws.

**Claims And Evidence:**

The claims of the authors are limited to the well-fittedness of their law-formulation to their empirical data. The data presented seem clear enough. The authors make scant claims of the effectiveness of their method compared to any baseline, and the claims they do make are not substantiated with experimentation, but rather exist as almost side-comments.

**Essential References Not Discussed:**

Not to this reviewers limited knowledge.

**Experimental Designs Or Analyses:**

This is a peculiar paper in that it does not really have experiments per se. The authors present empirical results on the setups they designed and show that their 'law' formulation is a good fit for the empirical data. But they do not apply this 'law' towards solving any extant articulated problem; there is no 'baseline' against which a novelty is compared. A stronger version of the paper might say something like "this allows us to more efficiently find CPT hyperparams which lead to good performance" and then show experiments against a baseline. Another potential experiment would be fitting the laws on some setup and extrapolating the fit laws to other setups, showing generalization across datasets or models.

**Methods And Evaluation Criteria:**

The paper does not really present baselines as it does not really present experiments. There are no figures or tables showing comparison to any other method. Rather, the paper presents empirical findings showing the proposed law fits nicely the data drawn from the various experimental setups.

**Other Comments Or Suggestions:**

The fundamental issue with this work is that it is a empirical study with neither theoretical analysis or demonstrated pragmatic value. The authors propose a law for continual pre-training loss evolution and show that this law fits nicely, but do not explicate or demonstrate why anyone might want to know about it, or how anyone might make use of it. The paper reads almost like an extended analysis section, missing experiments comparing to any baseline or a framing of the method which answers the question 'so what?'. For what the paper is, the analysis seems sound and the law seems to fit the data well. Whether or not this alone is sufficient for publication is dubious. It seems addressing the above concern could feasibly produce a much stronger presentation of this work.

**Other Strengths And Weaknesses:**

Clarity leaves much to be desired. The first of two "Research Questions" in the intro reads "can we find such an accurate law containing as many variables that affect the final CPT performance as possible?". This can be greatly improved. Notions 'loss potential' and 'turning length' are used before being explained in the introduction. Throughout, concepts are poorly articulated, making unclear in sections what the authors are attempting to communicate.

Significance is somewhat unclear. It may be interesting that the "laws" described fit the data nicely, but it is left unstated as to why this may be *useful*. There is not a clear statement of what problem may be solved by the description of the proposed "laws" and accordingly there is no experiment which demonstrates that problem as baseline and some methodological solution. Some of the "Findings" are of questionable value. #5 seems to be a series of uncontroversial statements which are so general as to be obvious.

Many works already exist fitting power laws to various aspects of machine learning, so originality of the meta-methodology is limited. The presented empirical analysis seems reasonable, if limited. The utility of the method is unclear.

**Questions For Authors:**

In the authors view, what is a practitioner who reads this work and is now famliar with the proposed 'laws' to do differently? What problem is being solved, and what evidence is there that is has been?

**Relation To Broader Scientific Literature:**

CPT is a currently topical method for analysis, as it is widely in use with LLMs. The 'laws' proposed in this work follow a general trend of identifying power laws in various machine learning settings to help make predictions about model performance, oftentimes with the goal of restricting the hyperparameter space of training and thus increasing the efficiency of producing useful models. This work evaluates laws which seem to fit nicely the data presented, though the contribution of those laws towards improving efficiency is left unstated and unexplored.

**Theoretical Claims:**

No theoretical claims present. The authors note this in limitations.

---

> ### Author Rebuttal · Authors · 2025-03-31
>
> We sincerely appreciate your suggestions.  We would also like to acknowledge the thorough and constructive feedback provided which help strengthen our work, which we summarize and respond as follows.
>
> **Response to "Our work is empirical":**
> Our work is indeed empirical. However, most scaling laws in LLMs are empirical, such as the OpenAI [1] scaling law and the Chinchilla [2] scaling law. We believe that it is common practice in developing scaling laws of LLMs.
> Moreover, our scaling law can be applied to many practical scenarios
> We have conducted extensive experiments to demonstrate the validity of our empirical law, including different learning rate schedules in Figure 3 and Figure 10, various model sizes in Appendix E, and different CPT datasets or replay ratios in Appendix D.
>
> **Response to "No baselines":**
> We are the first to propose the CPT law that traces the learning dynamics of LLMs throughout the CPT process and we model the full loss curve during CPT. Our CPT law considers numerous variables that affect CPT performance, which was not addressed in previous works. Consequently, we cannot compare the accuracy of fitting or prediction with other baselines due to the absence of comparable approaches. Nevertheless, to validate our CPT law, we have compared it with alternative law formulations as baselines, as detailed in Appendix I.
>
> **Response to "Contribution of laws", "Why this may be useful", "What problem may be solved", "No extant articulated problem" and "Questions: what is a practitioner who reads this work and is now familiar with the proposed 'laws' to do differently?":**
> Although our scaling law is empirical, it could help researchers understand the dynamics of continual pre-training, as noted by several reviewers. Reviewer nT6s states: "This work sets a stage for subsequent research for scaling to larger model scales during continual pre-training." Similarly, Reviewer 5BNQ observes: "Overall, this is an educational paper to understand how continual pre-training works," while Reviewer oHxM acknowledges: "These contributions are unique and not seen in prior works." Our scaling law especially assists researchers in understanding the impact of various hyperparameters on the complete learning dynamics. More specifically, our scaling law could fit by very few CPT data points and accurately predict the loss across diverse hyper-parameters (e.g. LR schedule) as shown in Figure 10, which enables to efficiently optimize the training hyperparameters (Figure 8) without extensive search procedures, thereby conserving computational resources.
>
> **Response to "Findings seems to be a series of uncontroversial statements which are so general as to be obvious":**
> We believe that some findings in our paper may not be so obvious to certain researchers. Our scaling law validates these obvious findings, which further demonstrates the correctness of our scaling law and reliability of these findings. Most importantly, our scaling law not only provides the qualitative analysis about these findings but also enables quantitative analysis for each finding. For example, it can predict the optimal loss potential, peak LR, and other hyperparameters through the scaling law as shown in Figure 8.
>
> **Response to "Loss Potential" and "Turning Length":**
> We apologize for any poor reading experience caused by organizational issues about these terms in our paper and will address these concerns in the final version.
> For loss potential, we define in line 187 "captures the potential of future loss drop via LR annealing.", which corresponds to the height in Figure 4(c). For example, if the pre-training implements substantial annealing and end at the valley bottom in Figure 4(c), the remaining annealing room or potential for CPT is diminished. Quantitatively, we define loss potential as the ratio of the final learning rate of the pre-training annealing phase to the initial or maximum learning rate in the pre-training phase: $ loss\ potential = \frac{Final\ LR\ of\ PT}{Maximum\ LR\ of\ PT} $, which is same as the legend in our Figure 5(b)(e) and x-axis of Figure 5(c)(f). In Figure 5(a)(d), if the pre-training model anneals from 2e-4 to 1e-4 using a linear method, the loss potential for this model is 50%.
> For turning length, we define it in line 315 as "The minimum training steps required to achieve a lower loss are designated as the turning length." We also illustrate the meaning of this term in Figure 7(c). Qualitatively, when the CPT steps once reach the turning length, the validation loss on pre-training dataset reverts to the same loss as the very beginning of CPT as shown in Figure 7(c).
>
>
> [1] Kaplan, Jared, et al. "Scaling laws for neural language models." arXiv preprint arXiv:2001.08361 (2020).
> [2] Hoffmann, Jordan, et al. "Training compute-optimal large language models." arXiv preprint arXiv:2203.15556 (2022).

---

> > ### Comment · Reviewer_oDDW · 2025-04-04
> >
> > Thanks for the thorough response. In light of the points made, I adjust my score upwards.

---

### Official Review · Reviewer_nT6s · 2025-03-25

**Overall Recommendation:** 4

**Summary:**

The paper explores learning dynamics in Continual Pre-Training (CPT) for LLMs, focusing on how general and downstream domain performance evolves at each training step, using validation losses. The paper observes that the CPT loss curve represents a transition between hidden loss curves, influenced by distribution shift and learning rate annealing. The paper empirically derives a CPT scaling law combining these factors, enabling loss prediction at any training step and across learning rate schedules in CPT. This scaling law sheds light on key CPT factors, including loss potential, learning rate, training steps, and replay ratio. The proposed approach enables customizing training hyper-parameters for different CPT goals, such as balancing general and domain-specific performance. Lastly, the paper extends the scaling laws to more complicated scenarios such as out-of-domain datasets and models with unknown pre-training details.

## update after rebuttal

Most of my concerns were addressed in the rebuttal response. I hope remaining clarity issues are fixed in the revision. Considering all feedback, I stand by my score and recommend acceptance.

**Claims And Evidence:**

The paper’s claims are well-supported by the provided evidence.

1. A CPT scaling law (Equation 4) is derived and validated by decomposing the CPT loss curve (Figures 1, 2). Experiments (Figure 3) demonstrate effectiveness for small models (0.1-1B parameters) across various training settings, though generalizability to larger models is a concern.

2. The paper analyzes key CPT factors (loss potential, distribution distance, learning rate, CPT steps) and their impact (Section 4, Figures 5-7).

3. Figure 8 demonstrates the approach’s adaptability for customizing hyperparameters to specific CPT goals, with more detailed discussions in Section 5.

4. The scaling law is shown to extend to OOD datasets (Subsection 5.2, Figure 9) and open-source models with unknown training details (Appendix G).

In summary, the paper provides substantial empirical support for its claims, including the empirical scaling laws. The analysis is detailed, and limitations regarding model scales are acknowledged.

**Essential References Not Discussed:**

The paper is organized well, citing all relevant prior work and detailing how the proposed work draws inspiration from and differs from existing research. To my knowledge, the paper does not omit any important related works or works that would hinder an understanding of the current paper.

**Experimental Designs Or Analyses:**

The paper generally employs sound experimental designs. The paper considers multiple setups with variations in training settings—learning rate schedules, model size, continual pre-training datasets, and hyperparameters—to test the robustness of the CPT scaling law. The paper uses established datasets (Fineweb, Knowledge-Pile) and standard evaluation metrics (validation loss). The analysis of the CPT loss curve and the validation of the scaling law through fitting empirical loss curves are appropriate (with details about the fitting procedure in Appendix C).

However, a key limitation is the lack of rigorous theoretical analysis and proof for the CPT scaling law, as acknowledged in the paper. The conclusions regarding model size are also based on assumptions, as the experiments did not reach the scale of current large language models and are limited to 0.1-1B parameter models.

**Methods And Evaluation Criteria:**

Based on the paper, the proposed methods and evaluation criteria make sense for the problem and application at hand. The reason being:

1. The paper analyzes and models learning dynamics in CPT of large language models, focusing on performance evolution across general and downstream domains during CPT, analyzing catastrophic forgetting.

2. The paper introduces a CPT scaling law combining distribution shift and learning rate annealing to predict loss at any training step and across learning rate schedules, modeling key CPT factors like loss potential, learning rate, training steps, and replay ratio.

3. The paper uses validation loss of corresponding domains to trace performance changes, a standard practice in language modeling and continual learning, and validates the scaling law using different learning rate schedules, datasets, and model sizes.

4. The paper uses standard datasets like FineWeb and Knowledge-Pile, enabling comparisons and ensuring relevance.

**Other Comments Or Suggestions:**

1. It would help to formally define “loss potential” for better readability and understanding of this introduced concept.

2. In Figure 5, there are few legends that are 0% loss potential while some are 10% loss potential. Is there a typo somewhere?

**Other Strengths And Weaknesses:**

The paper addresses the important and timely problem of understanding continual pre-training, a prevalent technique where a generic pre-trained model is further pre-trained with a domain-specific corpus for particular use cases. The paper is well-written, clearly positions itself within existing work, is easy to follow, and includes extensive experimentation to support the claims.

As discussed previously, two main weaknesses are the paper’s focus on smaller model scales (0.1-1B parameters), raising questions about the generality of findings to larger models, and the limited theoretical foundation for the derived scaling laws, as they are primarily empirical. However, the latter does not constitute a reason to reject the paper.

**Questions For Authors:**

1. What do 0-100% loss potential values mean? What is the baseline used to compute this percentage?

2. In Finding 3, the paper notes that “PT models with higher loss potential always achieve lower $D_{cpt}$ validation losses…”. However, "loss potential" is relative to $D_{pt}$ validation loss or $D_{cpt}$ validation loss. It is unclear from the context which loss is being referred to in this finding.

**Relation To Broader Scientific Literature:**

The paper analyzes and models the learning dynamics in Continual Pre-Training (CPT) of large language models, especially understanding how the performance of these models evolves across general and downstream domains during the domain-specific adaptation process. This is a critical issue in continual learning, as models often suffer from catastrophic forgetting, where they lose previously learned information when learning new tasks. So, identifying key operating points like optimal replay ratios, starting loss potentials, and learning rate schedules is very important. Although some of these details are well-known and intuitive, the paper takes a stab at defining them, deriving scaling laws for the same, and employing them even in more realistic scenarios like open-source pre-training language models with unknown training details. This work sets a stage for subsequent research for scaling to larger model scales during continual pre-training.

**Theoretical Claims:**

The paper derives scaling laws inspired by existing works and validates them by fitting all (continual) pre-training loss curves with different learning-rate schedules. So it is mainly an empirical paper.

---

> ### Author Rebuttal · Authors · 2025-03-31
>
> We sincerely appreciate your recognition of our work.
> **Response to "Limited theoretical foundation for the derived scaling laws":**
> Our work is indeed empircial. However, most scaling laws in LLMs are empirical, such as the OpenAI [1] scaling law and the Chinchilla [2] scaling law. We believe that it is common practice in developing scaling laws of LLMs.
>
> [1] Kaplan, Jared, et al. "Scaling laws for neural language models." arXiv preprint arXiv:2001.08361 (2020).
> [2] Hoffmann, Jordan, et al. "Training compute-optimal large language models." arXiv preprint arXiv:2203.15556 (2022).
>
> **Response to "Smaller model scales (0.1-1B parameters)":**
> The largerest model size in our experiment is 1.7B parameters without embedding, and we also conduct experiments with LLaMA-3.2-1B in Appendix G. We believe this model size is reansonable in the current context of large language models. Due to resource constraints, we are indeed unable to train larger models. Our scaling law has demonstrated consistent patterns across these model sizes (0.1-1.7B), so we believe our law is also applicable for larger models.
>
> **Response to "Comments Or Suggestions 1" and "Q1" about "Loss Potential":**
> We apologize for any poor reading experience caused by organizational issues about these terms in our paper and will address these concerns in the final version.
> For loss potential, we define in line 187 "captures the potential of future loss drop via LR annealing.", which corresponds to the height in Figure 4(c). For example, if the pre-training implements substantial annealing and end at the valley bottom in Figure 4(c), the remaining annealing room or potential for CPT is diminished. Quantitatively, we define loss potential as the ratio of the final learning rate of the pre-training annealing phase to the initial or maximum learning rate in the pre-training phase: $ loss\ potential = \frac{Final\ LR\ of\ PT}{Maximum\ LR\ of\ PT} $, which is same as the legend in our Figure 5(b)(e) and x-axis of Figure 5(c)(f). For example, in Figure 5(a)(d), if the pre-training model anneals from 2e-4 to 1e-4 using a linear method, the loss potential for this model is 50%.
>
> **Response to "Comments Or Suggestions 2: typo in figure 5":**
> We apologize for the typo in the legend of Figure 5(a); the correct value is 10% loss potential, not 0%. We will revise it in the final version.
>
> **Response to "Q2: loss potential is relative to D_pt validation loss or D_cpt validation loss":**
> Based on the defination of $loss\ potential$ above, the loss potential is neither related to D_PT validation loss nor D_CPT validation loss but rather represents the annealing potential of the models. Our findings indicate that pre-trained models with higher loss potential can achieve lower D_CPT validation losses, suggesting that these models could better adapt to the downstream datasets.

---

> > ### Comment · Reviewer_nT6s · 2025-04-09
> >
> > Thank you for your explanation. I hope the authors will resolve the clarity issues in future revisions. Having considered all the reviews and the authors' comments, I will maintain the current score.

---

### Decision · Program_Chairs · 2025-05-01

**Decision:**

Accept (oral)

**Comment:**

### Summary

This paper derives scaling laws for continual pretraining (CPT) scenarios, leveraging the work of Tissue et al (2024) to account for learning rate schedulings. They analysis the loss shift attributed to:
- distribution shift between PT and CPT domains
- change of learning rate (w/ and w/o decay, w/ and w/o warmup)

Important observations:
- the important finding that after a critical point, the moment where the PT->CPT transition is done has little to no impact, because there is a "hidden CPT training curve" that is reached
- the "turning length" for PT -> CPT transition done before critical point
- even when PT domain is not known, and generic PT dataset (e.g SlimPajama) can be used as a good proxy

New tool:
- the "loss potential" quantity that allow to predict how much improvement can be expected by further training, as function of last LR value
- for OOD domains, the law can be applied by finding coefficients lambda1 and lambda2 and express the new loss as linear combination of PT/CPT losses, regression lambda coefficients

### Rebuttal and discussion

Reviewers oHxM and oDDW mainly complained about a lack of clarity, to which authors answered in a satisfactory manner, and committed to improve the final version of the manuscript.

Reviewer nT6s noticed a lack of theoretical foundation, but said "However, the latter does not constitute a reason to reject the paper." to which I agree. They also noticed that the scale was limited to 1B model - which is already consequent.

Reviewer oDDW wrote: "The fundamental issue with this work is that it is an empirical study with neither theoretical analysis or demonstrated pragmatic value". I disagree and believed the authors did a good job in answering this point. The paper validates many intuitive findings, but also brings new insights, that have an obvious pedagogical value.

Besides that, the extensive experiments, the rigor, were all praised by reviewers.

### Recommendation

The paper has an obvious pedagogical value. Findings are new, timely, relevant. Experiments are rigorous. Message is conveyed in a clear manner.

If anything, the paper suffers from too many messages and findings, and some concepts were insufficiently explained (e.g turning length and loss potential); but the authors promised to improve the writing in the camera-ready version.

Overall, I would recommend the reading of this paper for anyone interested in continual pretraining.  Therefore I recommend acceptance.